# The skin microbiome facilitates adaptive tetrodotoxin production in poisonous newts

Patric M Vaelli[1,2†], Kevin R Theis[2,3], Janet E Williams[2,4,5], Lauren A O'Connell[6], James A Foster[2,5,7], Heather L Eisthen[1,2]*

[1]Department of Integrative Biology, Michigan State University, East Lansing, United States; [2]BEACON Center for the Study of Evolution in Action, Michigan State University, East Lansing, United States; [3]Department of Biochemistry, Microbiology, and Immunology, Wayne State University, Detroit, United States; [4]Department of Animal and Veterinary Science, University of Idaho, Moscow, United States; [5]Institute for Bioinformatics and Evolutionary Studies, University of Idaho, Moscow, United States; [6]Department of Biology, Stanford University, Stanford, United States; [7]Department of Biological Sciences, University of Idaho, Moscow, United States

*For correspondence:
eisthen@msu.edu

Present address: †Department of Molecular and Cellular Biology, Harvard University, Cambridge, United States

**Abstract** Rough-skinned newts (*Taricha granulosa*) use tetrodotoxin (TTX) to block voltage-gated sodium ($Na_v$) channels as a chemical defense against predation. Interestingly, newts exhibit extreme population-level variation in toxicity attributed to a coevolutionary arms race with TTX-resistant predatory snakes, but the source of TTX in newts is unknown. Here, we investigated whether symbiotic bacteria isolated from toxic newts could produce TTX. We characterized the skin-associated microbiota from a toxic and non-toxic population of newts and established pure cultures of isolated bacterial symbionts from toxic newts. We then screened bacterial culture media for TTX using LC-MS/MS and identified TTX-producing bacterial strains from four genera, including *Aeromonas*, *Pseudomonas*, *Shewanella*, and *Sphingopyxis*. Additionally, we sequenced the $Na_v$ channel gene family in toxic newts and found that newts expressed $Na_v$ channels with modified TTX binding sites, conferring extreme physiological resistance to TTX. This study highlights the complex interactions among adaptive physiology, animal-bacterial symbiosis, and ecological context.

## Introduction

Coevolutionary interactions among species are a central force driving the origin of novel, adaptive phenotypes, yet the traits under selection are often complex and arise from multifaceted interactions among genetic, physiological, and environmental forces that are not well understood (*Ehrlich and Raven, 1964*; *Futuyma and Agrawal, 2009*; *Schoener, 2011*). Chemical interactions among species in the form of defensive compounds have evolved across all domains of life, and these toxins often target evolutionarily conserved proteins in potential predators (*Brodie and Ridenhour, 2003*; *Hodgson, 2012*; *Whittaker and Feeny, 1971*). For example, tetrodotoxin (TTX), the primary neurotoxin found in poisonous pufferfishes (*Tsuda and Kawamura, 1952*), has been discovered across a broad phylogenetic distribution of animals (*Chau et al., 2011*; *Hanifin, 2010*). The unusual molecular structure of this toxin serves to selectively target voltage-gated sodium ($Na_v$) channels, which are critical for generating action potentials in neurons, muscles, and other excitable cells (*Hille, 2001*). Thus, TTX toxicity can have substantial impacts on eco-evolutionary interactions among species, impacting both the toxic species and potential predators.

**eLife digest** Rough-skinned newts produce tetrodotoxin or TTX, a deadly neurotoxin that is also present in some pufferfish, octopuses, crabs, starfish, flatworms, frogs, and toads. It remains a mystery why so many different creatures produce this toxin. One possibility is that TTX did not evolve in animals at all, but rather it is made by bacteria living on or in these creatures. In fact, scientists have already shown that TTX-producing bacteria supply pufferfish, octopus, and other animals with the toxin. However, it was not known where TTX in newts and other amphibians comes from.

TTX kills animals by blocking specialized ion channels and shutting down the signaling between neurons, but rough-skinned newts appear insensitive to this blockage, making it likely that they have evolved defenses against the toxin. Some garter snakes that feed on these newts have also evolved to become immune to the effects of TTX. If bacteria are the source of TTX in the newts, the emergence of newt-eating snakes resistant to TTX must be putting evolutionary pressure on both the newts and the bacteria to boost their anti-snake defenses. Learning more about these complex relationships will help scientists better understand both evolution and the role of beneficial bacteria.

Vaelli et al. have now shown that bacteria living on rough-skinned newts produce TTX. In the experiments, bacteria samples were collected from the skin of the newts and grown in the laboratory. Four different types of bacteria from the samples collected produced TTX. Next, Vaelli et al. looked at five genes that encode the channels normally affected by TTX in newts and found that all them have mutations that prevent them from being blocked by this deadly neurotoxin. This suggests that bacteria living on newts shape the evolution of genes critical to the animals' own survival.

Helpful bacteria living on and in animals have important effects on animals' physiology, health, and disease. But understanding these complex interactions is challenging. Rough-skinned newts provide an excellent model system for studying the effects of helpful bacteria living on animals. Vaelli et al. show that a single chemical produced by bacteria can impact diverse aspects of animal biology including physiology, the evolution of their genes, and their interactions with other creatures in their environment.

Rough-skinned newts (*Taricha granulosa*) are among the most poisonous TTX-producing animals and serve as an excellent model system for understanding ecological influences on toxin production and predation (*Figure 1A*). This species is endemic to the Pacific Northwest of North America, where certain populations possess high quantities of TTX relative to other TTX-laden species including pufferfishes and blue-ringed octopuses (*Hanifin, 2010*; *Williams, 2010*). In some populations, individual newts possess enough TTX to kill several adult humans (*Brodie et al., 2005*; *Hanifin, 2010*; *Hanifin et al., 1999*). Variation in newt toxicity is driven in part by the evolution of TTX resistance in predatory garter snakes (*Thamnophis sirtalis*), as TTX toxicity and resistance in newts and snakes are strongly correlated geographically, suggesting that these two phenotypes are coevolving (*Brodie et al., 2005*; *Brodie et al., 2002*; *Hanifin et al., 2008*). Furthermore, TTX resistant Na$_v$ channels have evolved independently across different garter snake populations, suggesting multiple independent origins of TTX resistance in predatory snakes (*Feldman et al., 2009*; *Geffeney, 2002*; *Geffeney et al., 2005*).

Despite the central role of TTX toxicity in coevolutionary interactions between newts and snakes, the origin of TTX in newts and other freshwater animals is unknown (*Daly, 2004*; *Hanifin, 2010*). In TTX-bearing marine species, toxicity is derived either from dietary accumulation from TTX-laden prey, or from symbiotic interactions with TTX-producing bacteria (*Chau et al., 2011*; *Miyazawa and Noguchi, 2001*). Pufferfishes harbor numerous TTX-producing bacteria symbionts in toxic tissues including skin, liver, intestines, and ovaries, and cultured non-toxic pufferfishes are able to sequester dietary-administered TTX under laboratory conditions (*Jal and Khora, 2015*). TTX-producing bacteria have also been isolated from xanthid crabs, horseshoe crabs, starfish, chaetognaths, nemerteans, gastropods, and blue-ringed octopuses (*Jal and Khora, 2015*; *Magarlamov et al., 2017*). However, the origin of TTX in rough-skinned newts has been more controversial (*Hanifin, 2010*). Newts raised in long-term captivity on artificial diets maintain their TTX toxicity (*Hanifin et al., 2002*), and captive

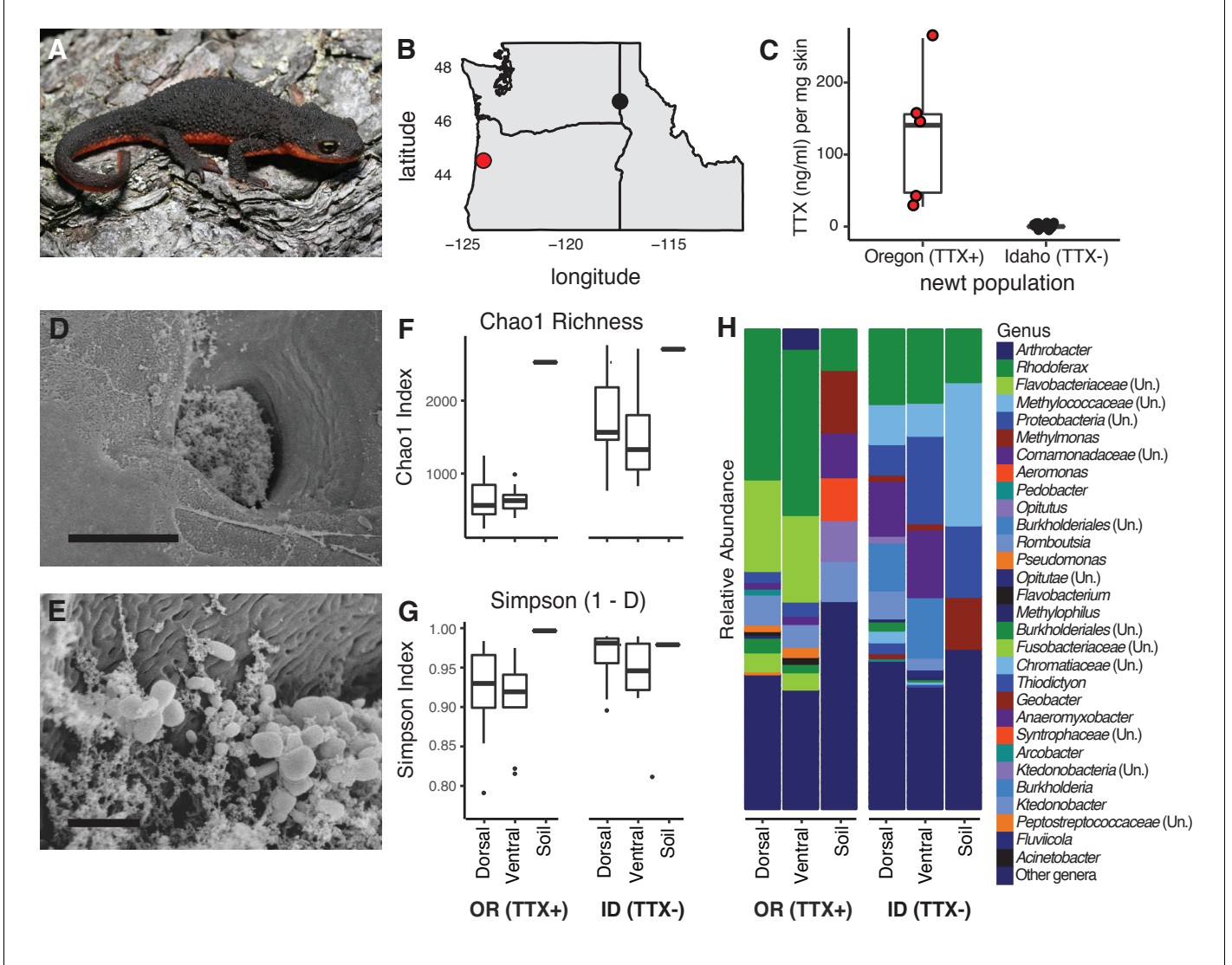

**Figure 1.** Characterization of the skin-associated microbiota of rough-skinned newts. (**A**) Rough-skinned newt (*Taricha granulosa*); photo by Gary Nafis (CC by ND NC 3.0). (**B**) Two populations of *T. granulosa* previously reported to possess either high concentrations of TTX (Lincoln Co, OR; red) or no TTX (Latah Co, ID; black) were compared in our study. (**C**) TTX measured from dorsal skin biopsies: newts from Oregon possessed 126.5 ± 42.1 ng mL$^{-1}$ TTX (n = 5) while Idaho newts possessed no detectable TTX (n = 17). (**D–E**) Scanning electron micrographs of host-associated bacterial communities upon the dorsal skin surface and within the ducts of TTX-sequestering granular glands. Scale bars 10 µm and 1 µm for D and E, respectively. (**F–G**) Comparison of bacterial community richness and diversity between these two populations, along with soil samples collected contemporaneously from the ponds in which the newts were caught. Non-toxic Idaho newts possessed higher OTU richness (Chao1 index, $t_{unequal\ var.}$ = 7.90, p<0.0001) and diversity (Simpson [1-D] index, $t_{unequal\ var.}$ = 4.11, p<0.0001) than toxic Oregon newts. (**H**) Mean relative abundance of bacterial OTUs present in dorsal or ventral skin of each population, as well as local soil samples. Sample sizes for panels F-H were n = 12 for Oregon (TTX+) and n = 16 for Idaho (TTX-) newts.

The online version of this article includes the following source data and figure supplement(s) for figure 1:

**Source data 1.** Raw data for newt skin toxicity measurements by LC-MS/MS.

**Source data 2.** The top 20 most abundant bacterial OTUs found among toxic (Oregon) and non-toxic (Idaho) newts.

**Source data 3.** Variation in alpha diversity between newt populations and sampling sites across the bodies of individual newts.

**Figure supplement 1.** The top 20 most abundant OTUs in the newt microbiome.

**Figure supplement 2.** Relative abundance of newt-associated bacteria at the phylum level.

newts forced to secrete their TTX by electric shock are able to slowly regenerate their toxicity over time (*Cardall et al., 2004*). Additionally, one group attempted to amplify bacterial DNA from various newt tissues using 16S rRNA gene primers, but failed to recover any PCR products aside from the

intestine, which contains low levels of TTX (*Lehman et al., 2004*). These studies suggest that the source of TTX in newts is not dietary, but whether newts have acquired the ability to produce TTX endogenously via convergent evolution or horizontal gene transfer, or from symbiosis with TTX-producing bacteria remains unclear.

Furthermore, despite the extreme toxicity of some newt populations, the molecular basis of TTX resistance in this species is not well understood. A previous study identified amino acid replacements in the highly conserved pore-loop (P-loop) region of the skeletal muscle isoform Na$_v$1.4 and found that skeletal muscle fibers were considerably resistant to TTX (*Hanifin and Gilly, 2015*). Amphibians, however, possess six Na$_v$ channel isoforms that are differentially expressed across excitable tissues (*Zakon et al., 2011*). Unlike pufferfishes in which TTX is sequestered in certain tissues, newts possess TTX throughout their bodies (*Mebs et al., 2010*; *Wakely et al., 1966*; *Yotsu et al., 1990*), indicating that the central and peripheral nervous systems are exposed to TTX. Thus, the evolution of whole animal resistance should necessarily involve all Na$_v$ channel subtypes, providing an opportunity to examine molecular evolution in response to a specific selective pressure (i.e., TTX) across an entire gene family.

In this study, we investigated the source of TTX toxicity and the molecular basis for TTX resistance in rough-skinned newts. We re-examined the hypothesis that newts derive their TTX from symbiosis with TTX-producing bacteria, focusing on the bacterial communities inhabiting the skin of *T. granulosa*, as this organ possesses specialized granular glands for storing toxins and contains the highest quantities of TTX in the animal (*Daly et al., 1987*; *Hamning et al., 2000*; *Hanifin et al., 2004*; *Santos et al., 2016*; *Toledo and Jared, 1995*; *Tsuruda et al., 2002*). We took advantage of the natural variation in TTX toxicity across newt populations to characterize the skin-associated microbiota in newts from a highly toxic and a non-toxic population and applied an unbiased cultivation-based approach to isolate numerous bacterial symbionts from the skin of toxic newts. Subsequent LC-MS/MS screening of bacterial cultivation media revealed TTX production in eleven bacterial strains from four genera: *Aeromonas*, *Pseudomonas*, *Shewanella*, and *Sphingopyxis*. Furthermore, to determine the molecular and physiological basis of extreme TTX resistance observed in this species, we cloned and sequenced five uncharacterized Na$_v$ channel paralogs (Na$_v$1.1, Na$_v$1.2, Na$_v$1.3, Na$_v$1.5, Na$_v$1.6) from a highly toxic population of newts in Oregon. We identified amino acid substitutions in all five genes, many of which have been observed in other TTX-possessing species. To test whether Na$_v$ channel mutations impact TTX resistance in newts, we used site-directed mutagenesis to insert three newt-specific replacements identified in Na$_v$1.6 into the TTX-sensitive Na$_v$1.6 ortholog from *Mus musculus*. We found that each amino acid replacement reduced TTX sensitivity compared to wild-type *M. musculus* channels, but these mutations had the greatest effect when combined together. Overall, our results suggest that host-associated bacteria may underlie the production of a critical defensive compound in a vertebrate host, impacting predator-prey coevolution and potentially shaping the evolution of TTX resistance in a well-known ecological model system.

## Results

### Characterization of newt-associated microbiota from toxic and non-toxic newts

To investigate whether bacterial symbionts produce TTX in newts, we leveraged natural variation in toxicity across newt populations to characterize skin-associated microbiota and determine whether highly toxic newts harbored candidate TTX-producing bacteria. We compared two populations previously reported to differ substantially in TTX levels (*Hanifin et al., 2008*; *Hanifin et al., 1999*), a toxic population in Lincoln County, OR and a non-toxic population in Latah County, ID (*Figure 1A–B*). As expected, skin biopsies collected from the dorsal skin of adult newts confirmed that Oregon newts (n = 5) possessed high TTX concentrations (126.5 ± 42.1 ng mL$^{-1}$ per mg skin) while Idaho newts (n = 17) lacked detectable levels of TTX (*Figure 1C* and *Figure 1—source data 1*). *T. granulosa* have particularly enlarged granular glands, which amphibians use to store and secrete noxious or toxic compounds (*Hanifin, 2010*; *Santos et al., 2016*; *Toledo and Jared, 1995*). Interestingly, scanning electron micrographs of dorsal granular glands revealed the presence of bacteria inhabiting the surface and outer pore of TTX-sequestering granular glands, and mixed bacterial communities including rod and coccus-shaped bacterial cells were present within the ducts of these glands

(*Figure 1D–E*). We therefore focused our sequencing and cultivation efforts on skin microbiota as a potential source of TTX in this species.

Bacterial communities inhabiting the dorsal and ventral skin, cloacal gland, and submandibular gland of *T. granulosa* were compared by culture-independent 16S rRNA gene sequencing targeting the V4 hypervariable region. Bacterial samples were collected by swabbing wild newts captured in the field at each site (Oregon n = 12 and Idaho n = 16). Bacterial samples were collected from the Oregon and Idaho populations at separate times, in June 2013 and September 2016, respectively. However, all bacterial DNA samples were extracted, amplified, and sequenced together on the same run. In total, we identified 4160 operational taxonomic units (OTUs) with an average Good's coverage (*Good, 1953*) of 0.9454 ± 0.0067 (mean ± SEM) across samples from both populations. 614 OTUs were unique to toxic newts, 1943 were unique to non-toxic newts, and 1603 were shared between the two populations. Among the 20 most abundant OTUs, 8 OTUs were shared between both populations while 12 were present in only one population (*Figure 1—figure supplement 1* and *Figure 1—source data 2*). These highly abundant and conserved OTUs may represent core skin microbiota of *T. granulosa*. Idaho newts possessed a greater number of distinct bacterial types with a more even distribution across their microbiota, reflected in a higher number of observed OTUs ($t_{unequal\ var.}$ = 7.70, p<0.0001) and higher OTU richness (Chao1 index, $t_{unequal\ var.}$ = 7.90, p<0.0001) on average than in Oregon newts. Bacterial alpha diversity was also significantly greater in Idaho than Oregon newts (Simpson 1-D index [$t_{unequal\ var.}$ = 4.11, p<0.0001]) (*Figure 1F–G* and *Figure 1—source data 3*). Phylum-level divisions show that newt microbiota consists primarily of Proteobacteria, Bacteroidetes, and Firmicutes, which together comprise 76.2–83.5% of the average bacterial community across all four body sites in both populations (*Figure 1—figure supplement 2*). At the genus level, the relative abundance of each bacterial OTU differed markedly between the two populations, as well as from soil samples collected from their respective habitats (*Figure 1H*).

The composition and relative abundances of OTUs (i.e. beta diversity) also differed significantly between the two newt populations (*Figure 2*). Principal coordinates analysis (PCoA) shows a distinct clustering based on geographic location in both composition (Jaccard index) and structure (Bray-Curtis index) of skin microbiota from each population (*Figure 2A–B*). Permutational multivariate analysis of variance (PERMANOVA) tests revealed that different skin sites across the animal harbored similar communities within a population (Jaccard, p=0.375; Bray-Curtis, p=0.065), but that community composition (Jaccard index, $F$ = 18.12, p<0.0001) and structure (Bray-Curtis index, $F$ = 40.40, p<0.0001) differed significantly between populations. The skin communities of Idaho newts were also more variable than those of Oregon newts (Permutational test for multivariate dispersion, PERMDISP, p=0.0053) (*Figure 2—figure supplement 1*). Interestingly, toxic Oregon newts maintain a high relative abundance of *Pseudomonas* OTUs relative to non-toxic Idaho newts (*Figure 2C*). Three *Pseudomonas* OTUs (00042, 00224, and 00485) were present in greater relative abundance in toxic newts, and OTU00042 was a significant driver of the beta diversity differences observed between these two populations. Indeed, linear discriminant analysis effect size (LEfSE) indicates that *Pseudomonas* OTU00042 is among the top 10 most differentially abundant OTUs in Oregon newts (*Figure 2D*). In subsequent non-targeted cultivation of newt skin bacteria, we isolated numerous culturable TTX-producing *Pseudomonas* spp. strains from toxic newts (below).

## Culturable newt microbiome and TTX production in bacterial isolates

To determine whether newt skin microbes produce TTX, we employed an unbiased, cultivation-based small molecule screen to examine bacterial culture media for the presence of TTX production in vitro. This approach was necessary because the genetic basis of TTX biosynthesis is unknown, preventing application of metagenomic or other sequencing approaches to determine whether newts or their microbiota possess the genetic toolkit for TTX production. Mixed bacterial communities were collected by swabbing the dorsal skin of toxic newts and cultured on nutrient-limited minimal media (Reasoner's 2A agar) or blood agar supplemented with defibrinated sheep's blood (10% v/v). Individual colonies were re-streaked, isolated in pure culture, and taxonomically identified by 16S rRNA gene sequencing. In total, we generated a culture collection of 355 strains representing 65 bacterial genera (summarized in *Figure 3—figure supplement 1*). Isolated strains were subsequently grown in 10% Reasoner's 2 broth (R2B) for 2 weeks at 20 °C; 1 mL of each culture was then removed and examined for TTX using mixed cation exchange solid-phase extraction (SPE) for sample

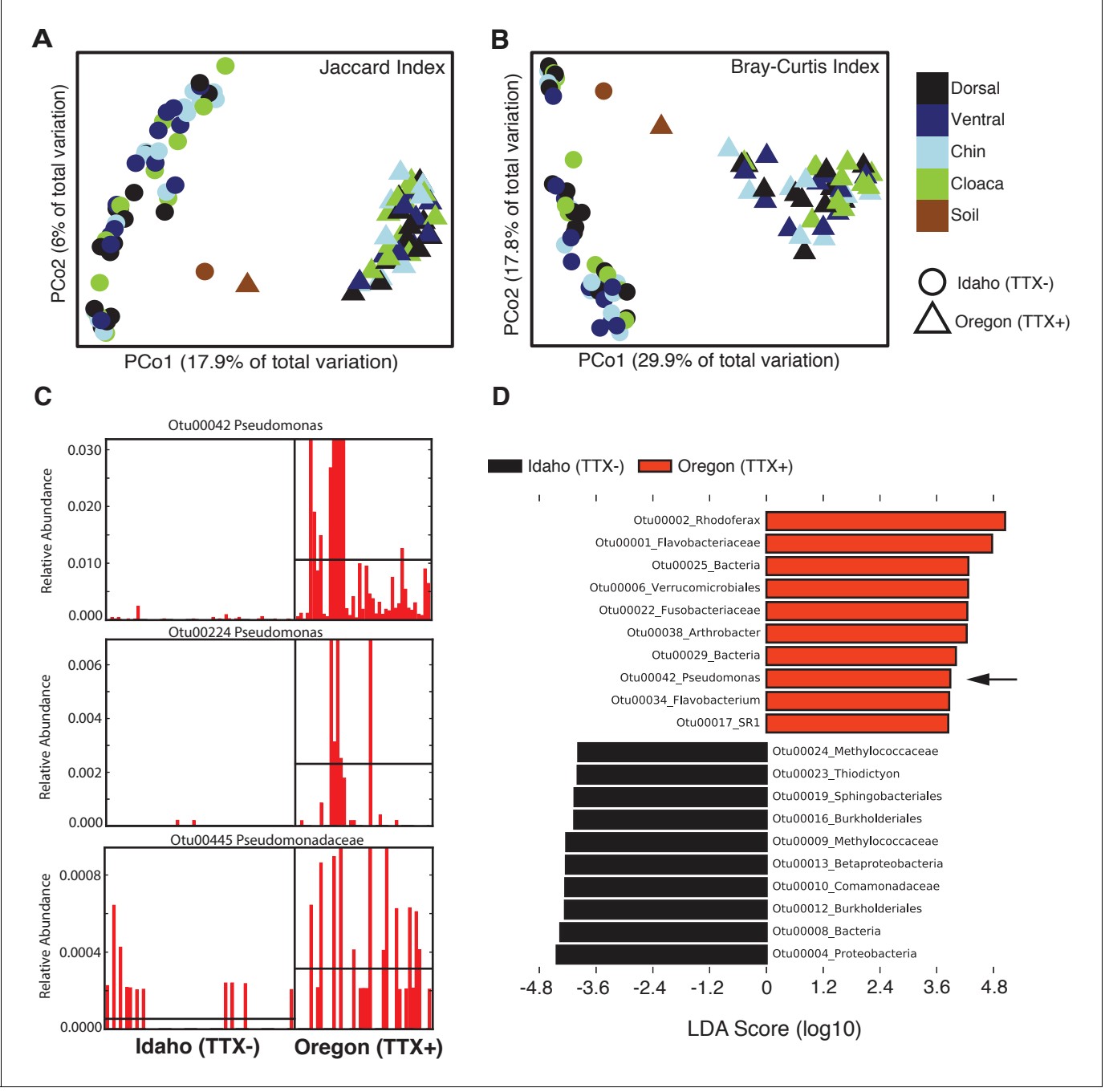

**Figure 2.** Comparison of skin microbiota from toxic and non-toxic newt populations reveal distinct host-associated bacterial communities and enrichment of TTX-producing bacteria. (**A–B**) Principal coordinates analysis of bacterial community composition (OTU presence/absence; Jaccard index) and community structure (OTU relative abundance; Bray-Curtis index) of skin microbiota across four body sites from toxic Oregon and non-toxic Idaho newts reveal distinct clustering of bacterial communities within each population. Non-parametric multivariate analysis of variance (PERMANOVA) test indicates a significant effect of location on both the composition (*F* = 18.12, p<0.0001) and structure (*F* = 40.40, p<0.0001) of skin-associated bacterial communities. Newt-associated communities also cluster separately from soil communities sampled from each location. (**C**) Representative comparison of highly abundant OTUs from individual toxic Oregon and non-toxic Idaho newt microbiota samples (on x-axis) reveals increased relative abundance of *Pseudomonas* OTUs in toxic newts. (**D**) Linear discriminant analysis effect size (LEfSE) comparing the top 10 differentially abundant OTUs between toxic and non-toxic newt microbiota indicates that *Pseudomonas* OTU00042 is highly enriched in toxic newt microbiota (black arrow). *Pseudomonas* isolated from newt skin produced TTX (*Figure 3*). Sample sizes were n = 12 and n = 16 for Oregon and Idaho newts, respectively.

The online version of this article includes the following figure supplement(s) for figure 2:

*Figure 2 continued on next page*

*Figure 2 continued*

**Figure supplement 1.** Principal coordinates analysis highlighting variation in bacterial community structure (Bray-Curtis index) of Oregon (red circles) and Idaho (black squares) newt skin microbiota.

**Figure supplement 2.** Relative abundance of all OTUs classified to the same genera as TTX-producing bacteria identified in this study (*Aeromonas, Pseudomonas, Shewanella,* and *Sphingopyxis*).

purification and liquid chromatography tandem mass spectrometry (LC-MS/MS) for molecular screening (*Figure 3A*).

Applying this strategy, we detected TTX in cultures from four distinct genera: *Aeromonas, Pseudomonas, Shewanella,* and *Sphingopyxis* (*Table 1* and *Figure 3B*). LC-MS/MS screening confirmed the presence of product ions at 162.1 and 302.1 m/z, corresponding to the primary and secondary product ions formed by TTX fragmentation, respectively (*Jen et al., 2008*). For quantification, we ran standard curves of 0.01, 0.05, 0.1, 0.5, 1, 2.5, 5, 10, and 25 ng mL$^{-1}$ pure TTX and used linear regression to estimate TTX concentrations in bacterial culture media from the base peak intensity (BPI) signal in each LC-MS/MS run. Across all TTX-positive samples, TTX concentrations produced were on average $0.236 \pm 0.087$ ng mL$^{-1}$ (mean $\pm$ SEM, n = 14). TTX was occasionally detected in samples with signals clearly above background noise and our limit of detection (LOD), but below our lower limit of quantification (LLOQ; 0.01 ng mL$^{-1}$). These samples were not included in our quantitative analyses, but this observation suggests that TTX production could be enhanced with strain-specific optimization of bacterial culture conditions.

TTX was detected in seven independent isolates of *Pseudomonas* spp. cultured from newt skin. Pairwise alignment of 16S rRNA gene sequences suggests that these isolates may represent four bacterial strains (*Figure 3—figure supplement 2*). Strains TX111003, TX174011, and TX180010 shared >99% nucleotide identity across homologous bases, and TX135003 and TX135004 also shared >99% sequence identity, but these two groups appeared to be distinct from each other (maximum similarity is 96.1%); these two groups were also isolated on different cultivation media, blood agar and R2A agar, respectively. 16S rRNA gene sequences for the remaining two *Pseudomonas* spp., strains TX111008 and TX111009, were unique from each other and the other two groups. Furthermore, we identified two TTX-producing *Shewanella* spp. strains that were isolated on different media and shared 94.2% 16S sequence identity. Thus, it appears several strains of *Pseudomonas* and *Shewanella* can produce TTX in lab culture. We also identified one individual strain each of *Aeromonas* and *Sphingopyxis* that produced TTX under these culture conditions (*Table 1*). Identification of numerous TTX-producing symbionts from distinct genera present on newt skin is consistent with observations in other toxic animal hosts such as the pufferfish and blue-ringed octopus, from which numerous distinct TTX-producing strains have been isolated (*Magarlamov et al., 2017*). However, we note that the vast majority of bacterial isolates screened in this study did not produce TTX under our culture conditions.

## Molecular basis of TTX resistance in newt Na$_v$ channels

Rough-skinned newts are the most toxic of TTX-producing animals (*Hanifin, 2010*), but the molecular basis of their TTX resistance has not been characterized. To determine the basis of TTX resistance in *T. granulosa*, we sequenced the Na$_v$ channel gene family and investigated the TTX-binding site, the S5-6 pore loop (P-loop), to determine if they possessed adaptive mutations that affect TTX binding and resistance. We generated transcriptomes from two excitable tissues (brain and nose) from a toxic newt and obtained partial sequences for five *SCN* genes that encode Na$_v$1.1, Na$_v$1.2, Na$_v$1.3, Na$_v$1.5, and Na$_v$1.6 proteins. We then cloned and sequenced the DI-DIV transmembrane sequences of each gene for verification, including the Na$_v$1.6 channel of both toxic and non-toxic newts. *SCN4A* (Na$_v$1.4) was obtained from GenBank for sequence comparison (accession number KP118969.1).

Although S5-S6 pore-loop (P-loop) sequences are highly conserved across the vertebrate Na$_v$ gene family, several amino acid substitutions were present in the P-loops across all six Na$_v$ channels in *T. granulosa* (*Figure 4*). In DI, Tyr-371 was replaced independently across four of the six channels; this parallel substitution involves a replacement from an aromatic Tyr or Phe to a non-aromatic amino acid, either Cys, Ser, or Ala. In the mammalian Na$_v$1.5 channel, this site is also replaced with a

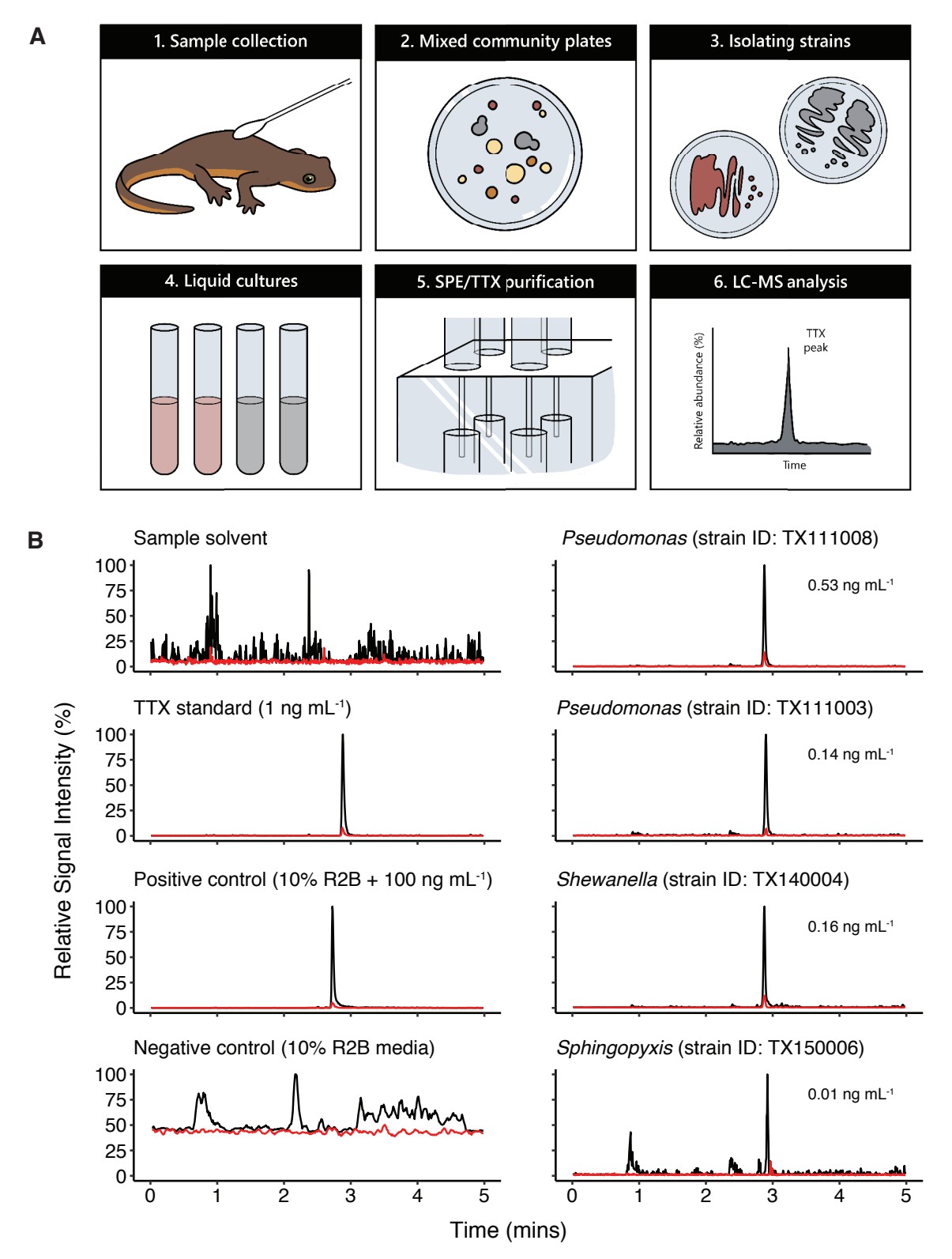

**Figure 3.** Bacterial isolates cultured from newt skin produce TTX in vitro. (**A**) Schematic overview of procedure for isolating and screening newt bacteria for TTX production. Bacterial samples were collected from dorsal skin of toxic newts, taxonomically identified by 16S rRNA gene sequencing, grown in liquid culture for 2 weeks, centrifuged, and the supernatant was purified by solid-phase extraction (SPE). Extracts were screened against TTX analytical standards by LC-MS/MS. (**B**) Representative extracted ion chromatographs showing peaks corresponding to major product ion transitions 320.1 to 162.1

*Figure 3 continued on next page*

*Figure 3 continued*

*m/z* (black) and 320.1 to 302.1 *m/z* (red) for TTX in cultures from four bacterial isolates. The retention time for each peak was 2.9 min and matched that of both authentic TTX standards and culture media supplemented with 100 ng mL$^{-1}$ TTX. Peaks were absent in untreated culture media. Estimated TTX concentrations in bacterial cultures are shown next to each peak. Multiple strains of *Pseudomonas* spp. were found to produce TTX and three additional TTX-producing genera were identified: *Aeromonas*, *Shewanella*, and *Sphingopyxis*.

The online version of this article includes the following figure supplement(s) for figure 3:

**Figure supplement 1.** Phylogenetic tree of bacteria cultivated from the skin of toxic rough-skinned newts.

**Figure supplement 2.** Pairwise comparison of 16S rRNA sequences between TTX-producing strains of *Pseudomonas* spp. identified in this study.

Cys and has been shown to underlie the classic TTX resistance of the cardiac Na$^+$ current (*Satin et al., 1992*). An additional DI difference was found at N374T in Na$_v$1.5; this site is also altered in amniotes, but not in *Xenopus* frogs, indicating that these mutations may be convergent. In DII, only one substitution is present at T938S in Na$_v$1.3, which is adjacent to the electronegative Glu-937 that directly binds the positively charged guanidinium group of TTX (*Shen et al., 2018*). This region is otherwise well-conserved across tetrapods, suggesting that the DII P-loop sequence is under strong purifying selection. Three sites differed in DIII including V1407I, M1414T, and A1419P in Na$_v$1.6, Na$_v$1.4, and Na$_v$1.1, respectively. The DIII M1414T substitution is present in at least five Na$_v$ channel paralogs in TTX-laden pufferfishes and increases TTX resistance at least 15-fold (*Jost et al., 2008*); thus, *T. granulosa* and pufferfish have converged on the identical molecular solution to reduce TTX sensitivity. The other two differences have not been previously characterized, but we subsequently tested the effects of Na$_v$1.6 V1407I on TTX binding (below). Finally, replacements in DIV occur across four sites, including a substitution of A1703G in the selectivity filter DEKA motif in Na$_v$1.2, Ile-1699 in Na$_v$1.4 and Na$_v$1.6, D1706S in Na$_v$1.4, and Gly-1707 in Na$_v$1.1, Na$_v$1.2, and Na$_v$1.4.

To determine whether TTX resistance was conferred by the P-loop mutations in *T. granulosa*, we focused on the neural subtype Na$_v$1.6, which is widely expressed in both the central and peripheral nervous system (*Caldwell et al., 2000*; *Hu et al., 2009*; *Lorincz and Nusser, 2010*; *Mercer et al., 2007*). We identified three amino acid replacements in the Na$_v$1.6 channel of both toxic and non-toxic newts (*Figure 5A*) and used site-directed mutagenesis to insert each mutation (DI Y371A, DIII V1407I, and DIV I1699V), as well as all three mutations, into the TTX-sensitive Na$_v$1.6 ortholog from mouse. We found that TTX sensitivity was greatly reduced in triple mutant channels (*Figure 5B–C*), and that each individual substitution contributed to TTX resistance (*Figure 5—figure supplement 1*). Estimated half-maximal inhibitory concentrations (IC$_{50}$) confirmed that DIII and DIV mutations provided 1.5-fold and 3-fold increases in resistance, respectively, while the DI and triple mutant channels were estimated to provide a > 600 fold increase in resistance (*Table 2* and *Figure 5D*).

**Table 1.** Summary of TTX-producing bacteria isolated from rough-skinned newts.

Genus-level identification of each TTX-producing isolate was determined by 16S rRNA gene sequencing and taxonomic classification by the Ribosomal Database Project classifier tool at 80% similarity cut-off (*Cole et al., 2014*). TTX production was determined by screening 1 mL culture media using LC-MS/MS. Overall, we cultured 11 TTX-producing isolates, though 16S gene sequence alignments suggest that some isolates may represent the same bacterial strain (see text). In most cases, TTX was detected in replicate cultures above the limit of detection (LOD), but not always above the lower limit of quantification (LLOQ). Mean TTX production ± standard error (SEM) is shown for samples above the LLOQ. Some of these bacterial genera contain TTX-producing strains that have previously been identified in other toxic animals (reviewed in *Chau et al., 2011*).

| Genus | Isolation media | TTX-producing isolates | Replicates above LOD | Replicates above LLOQ | TTX (ng mL$^{-1}$)± SEM | TTX symbiont in other animals |
|---|---|---|---|---|---|---|
| *Aeromonas* | Blood agar | 1 | 1 | 1 | 0.12 | Pufferfishes, sea snails |
| *Pseudomonas* | R2A or blood agar | 7 | 23 | 9 | 0.19 ± 0.07 | Pufferfishes, Blue-ringed octopus, sea snails |
| *Shewanella* | R2A or blood agar | 2 | 4 | 3 | 0.49 ± 0.36 | Pufferfishes |
| *Sphingopyxis* | R2A | 1 | 2 | 1 | 0.01 | None |

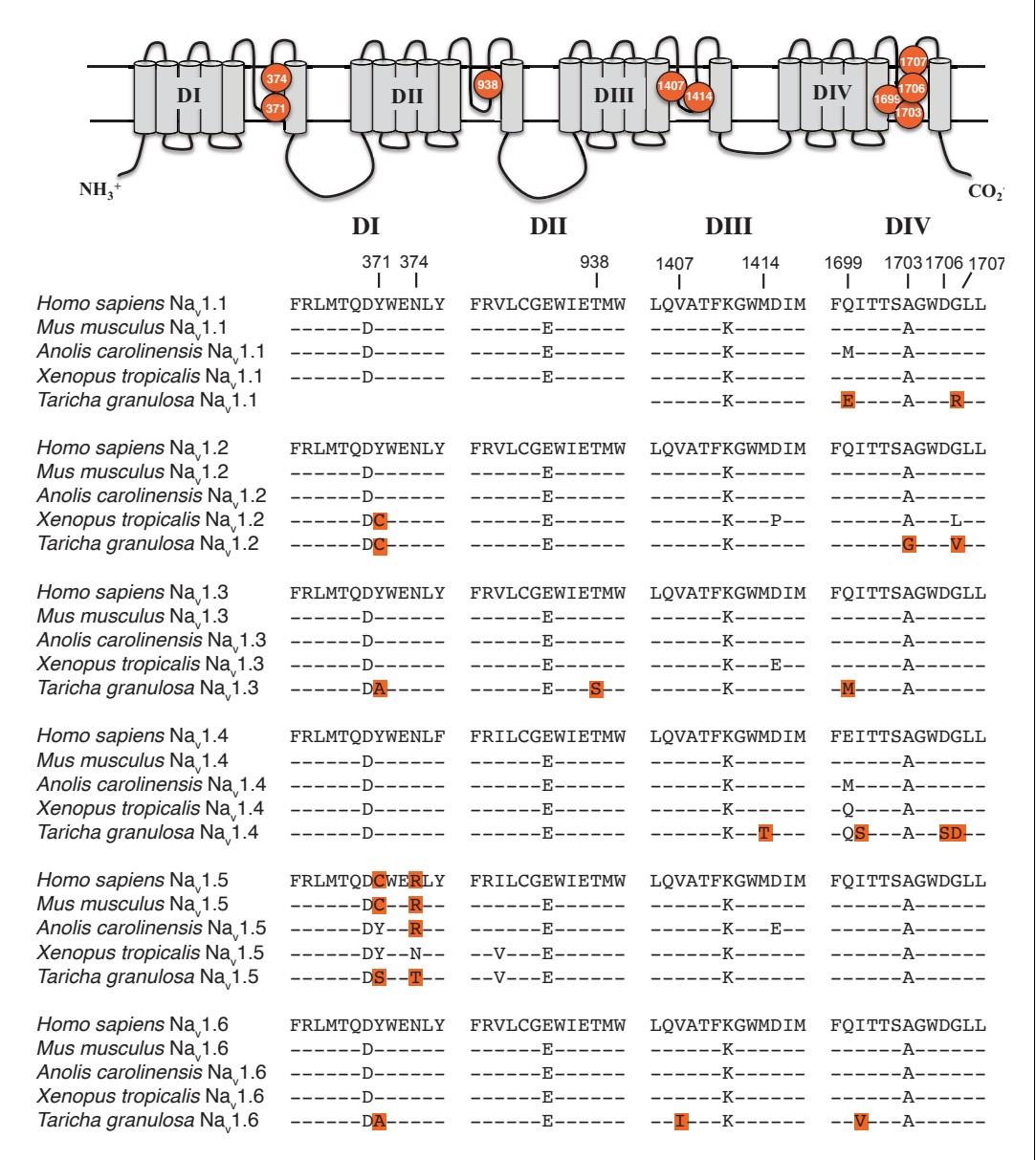

**Figure 4.** Protein alignment of Na$_v$ channels across representative vertebrates. Sequence alignment of S5-S6 P-loops from newts and other vertebrates showing amino acid substitutions relative to the P-loop consensus sequence for each Na$_v$ channel shown here. Putative TTX resistance mutations are highlighted in orange; mutations that are not highlighted are either synapomorphic in a gene clade or are present in TTX sensitive channels. Data are missing for DI and DII of Na$_v$1.1 in newts, which we did not recover in our sequencing efforts. The approximate locations of newt mutations are shown as orange circles, and the amino acid site of each mutation is numbered based on Na$_v$1.6 from *Mus musculus*.

The online version of this article includes the following source data and figure supplement(s) for figure 4:

**Source data 1.** GenBank accession numbers of vertebrate Na$_v$ channel protein sequences used in multiple sequence alignments and analysis.
**Figure supplement 1.** Parallel evolution of DIII and DIV P-loop substitutions in Na$_v$1.6 of toxic newts and TTX resistant garter snakes.

Thus, while all three mutations impact TTX resistance, the DI Y371A replacement provides considerable resistance independently. These results show that the three P-loop modifications in newt Na$_v$1.6 provide resistance to even extremely high concentrations of TTX, and comparison of Na$_v$ sequences from toxic and non-toxic newts revealed identical substitutions in both populations, suggesting that newts are broadly TTX-resistant regardless of toxicity.

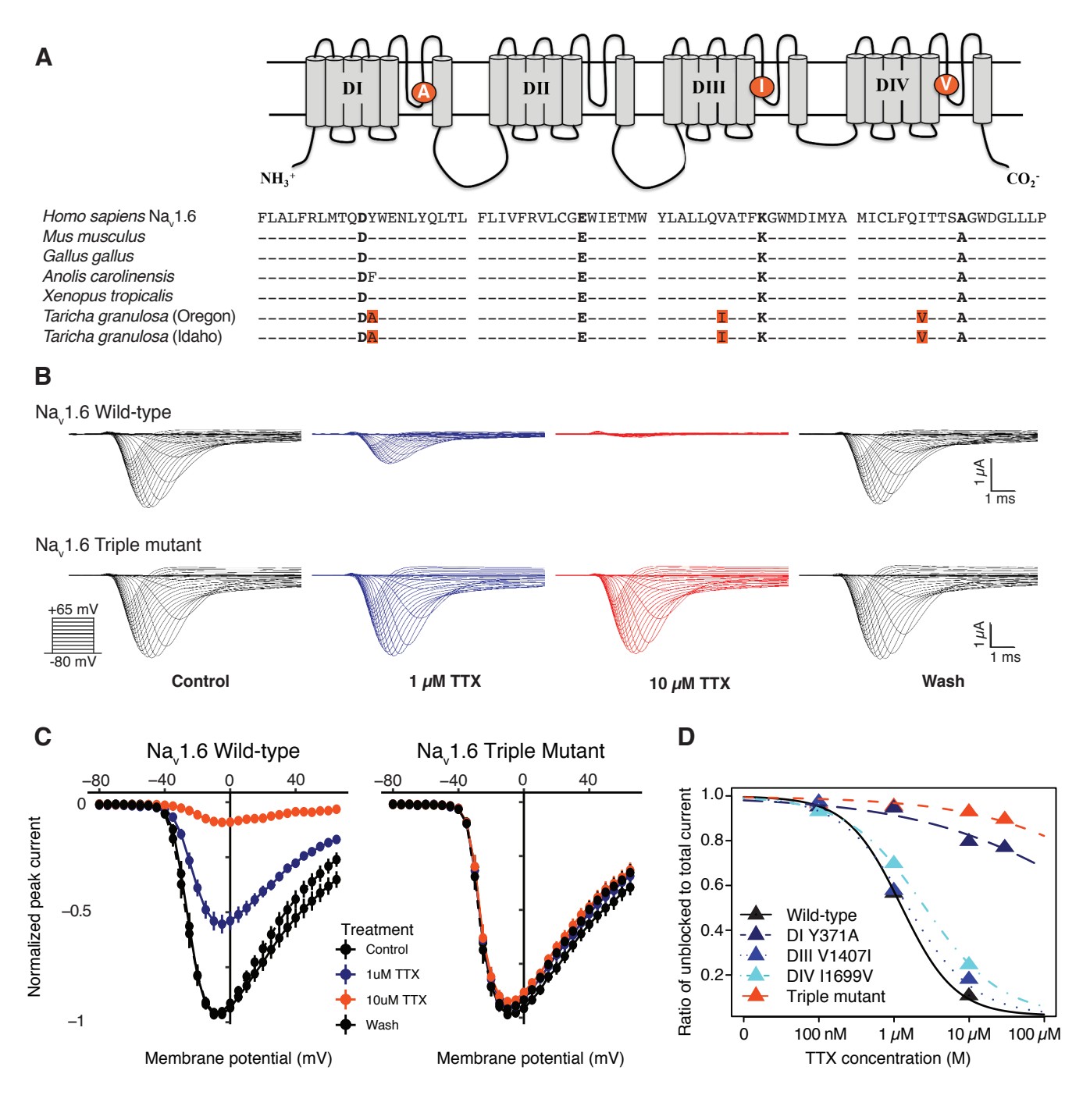

**Figure 5.** Newts possess Na$_v$ channel mutations that confer physiological resistance to TTX. (**A**) Predicted topology of Na$_v$1.6 with mutations in domains I, III, and IV. Sequence alignment of Na$_v$1.6 pore-loop motifs revealed three amino acid differences in newts from Oregon or Idaho populations. (**B**) Representative currents from wild-type mouse Na$_v$1.6 or Na$_v$1.6 with newt substitutions Y371A, V1407I, and I1699V treated with 1 μM (blue) or 10 μM (orange) TTX. (**C**) Current-voltage (I–V) relationships showing normalized currents for wild-type (n = 21) and mutant Na$_v$1.6 (n = 20) channels. Wild-type Na$_v$1.6 was blocked by TTX (Tukey's multiple comparisons test with Bonferroni correction: control vs. 1 μM, p<0.0001; control vs. 10 μM, p<0.0001), while mutated Na$_v$1.6 was unaffected (repeated measures ANOVA, p=0.879). (**D**) Dose-response curves showing the proportion of Na$^+$ current elicited during a step depolarization from −100 to −20 mV for wild-type, individual mutants, and triple-mutant Na$_v$1.6 channels exposed to increasing concentrations of TTX. Sample sizes are provided in *Table 2*. Data were fit with a Hill equation to estimate IC$_{50}$ values.

The online version of this article includes the following figure supplement(s) for figure 5:

*Figure 5 continued on next page*

*Figure 5 continued*

**Figure supplement 1.** Newt Na$_v$1.6 mutations increase TTX resistance in the orthologous mouse Na$_v$1.6.

## Discussion

In this study, we found that bacterial isolates from four genera, *Aeromonas, Pseudomonas, Shewanella*, and *Sphingopyxis*, cultured from the skin of *T. granulosa* produce TTX under laboratory conditions. Although TTX-producing symbionts have been identified in marine animals (*Chau et al., 2011*), this is the first identification of TTX-producing bacteria associated with a freshwater or terrestrial animal. The origin of TTX in rough-skinned newts and other amphibians has been controversial: wild-caught toxic newts maintain their toxicity in long-term laboratory captivity (*Hanifin et al., 2002*), and newts forced to secrete their TTX by electric shock regenerate their toxicity after nine months, despite laboratory conditions that prevented access to dietary sources of TTX (*Cardall et al., 2004*). Such results demonstrate that newts do not derive TTX from their natural diet, but the results do not explicitly rule out a symbiotic origin for TTX toxicity. A subsequent investigation attempted to amplify 16S rRNA genes from DNA extracted from newt tissues by PCR, but the authors were unable to amplify bacterial DNA from any tissue except the gut (*Lehman et al., 2004*). This result has been widely cited to claim that newts lack symbiotic bacteria altogether, thus supporting an endogenous origin for TTX (*Cardall et al., 2004*; *Gall et al., 2011*; *Gall et al., 2014*; *Hanifin, 2010*; *Hanifin and Gilly, 2015*; *Williams, 2010*). However, sequencing-based approaches for characterization of microbial communities were limited at that time, and it is increasingly clear that most, if not all, animals possess cutaneous bacterial communities on their external epithelium (*McFall-Ngai et al., 2013*). Thus, our results strongly suggest that symbiotic bacteria are the ultimate source TTX toxicity in rough-skinned newts.

Surprisingly, many of the TTX-producing strains isolated from newts are from the same genera as those previously identified in marine animals. TTX-producing *Pseudomonas* spp. have been isolated from toxic pufferfish, blue-ringed octopus, and sea snails (*Cheng et al., 1995*; *Hwang et al., 1989*; *Yotsu et al., 1987*), and TTX-producing *Aeromonas* spp. and *Shewanella* spp. have both been isolated from pufferfish and sea snails (*Auawithoothij and Noomhorm, 2012*; *Cheng et al., 1995*; *Simidu et al., 1990*; *Wang et al., 2008*; *Yang et al., 2010*). TTX-producing *Sphingopyxis* spp. have not been identified in host animals or environmental samples, and this strain may be unique to freshwater or terrestrial environments. Interestingly, several other newt species from diverse genera are known to possess TTX, including *Notophthalmus, Triturus*, *Cynops, Paramesotriton, Pachytriton*, and *Laotriton* (*Brodie et al., 1974*; *Yotsu-Yamashita and Mebs, 2001*; *Yotsu-Yamashita et al., 2007*; *Yotsu-Yamashita et al., 2017*). Frogs and toads from the genera *Atelopus, Brachycephalus, Colostethus*, and *Polypedates* also possess TTX (*Daly et al., 1994*; *Kim et al., 2003*; *Mebs et al., 1995*; *Tanu et al., 2001*; *Yotsu-Yamashita and Tateki, 2010*), as well as two species of freshwater flatworms (*Stokes et al., 2014*). Thus, the TTX toxicity observed in other amphibians and freshwater animals could be derived from bacterial sources similar to those identified in this study.

One of the most interesting insights to arise from this work is the possibility that the skin microbiome contributes to the predator-prey arms race between toxic newts and TTX-resistant garter

**Table 2.** Estimated half-maximal inhibitory concentrations (IC$_{50}$) of TTX for each Na$_v$1.6 construct. IC$_{50}$ values are shown as the concentration (mean ± SEM) of TTX (µM) that blocked half of the channels, estimated from the dose-response curve. The IC$_{50}$ ratio was taken as the fold increase in TTX resistance.

| Construct | N | IC$_{50}$ | IC$_{50}$ Ratio |
|---|---|---|---|
| Mouse Na$_v$1.6 | 21 | 1.25 ± 0.09 | 1 |
| DI (Y371A) | 17 | 763.7 ± 284 | 609 |
| DIII (V1407I) | 13 | 2.34 ± 0.23 | 1.2 |
| DIV (I1699V) | 15 | 4.73 ± 0.42 | 2.0 |
| Triple mutant | 20 | 3551 ± 469 | 2832.4 |

snakes. Populations of garter snakes sympatric with TTX-laden newts possess several amino acid replacements in their Na$_v$ channels that prevent TTX binding, allowing resistant snakes to prey on highly toxic newts (*Feldman et al., 2009*; *Geffeney, 2002*; *Geffeney et al., 2005*). As snake populations accumulate stepwise adaptive mutations in their Na$_v$ channels, selection drives increasing levels of toxicity in newts. Reciprocal selection for elevated toxicity and resistance in newt and snake populations, respectively, leads to an asymmetric escalation of these two traits, or a 'coevolutionary arms race' (*Brodie and Brodie, 1999*; *Brodie et al., 2005*; *Dawkins and Krebs, 1979*). If selection by predatory garter snakes favors increasing levels of toxicity in newt populations, selection may be acting not only on genetic variation in the host species, but also potentially on variation across the skin microbiome.

Selection could also act by increasing the relative abundance of TTX-producing symbionts in the skin (*Bordenstein and Theis, 2015*; *Theis et al., 2016*). Consistent with this hypothesis, we found that three abundant *Pseudomonas* OTUs were present in greater relative abundance in the microbiota of toxic newts compared to non-toxic newts (*Figure 2C–D*). *Pseudomonas* OTU00042 was particularly abundant in toxic newts and a significant driver of beta diversity between the toxic and non-toxic populations. Numerous TTX-producing *Pseudomonas* strains were also isolated in our cultivation assay, suggesting that this differential abundance may contribute to observed variation in TTX toxicity across newt populations. However, we did not observe a differential abundance of *Aeromonas* OTUs, which were abundant in both populations, nor of *Shewanella* or *Sphingopyxis* OTUs, which were found only on toxic newts, but were only present in a few samples and in very low abundance (*Figure 2—figure supplement 2*). These results may also reflect more favorable culture conditions for TTX-producing *Pseudomonas* spp. than for the other genera. Thus, further population-level comparisons across toxic and non-toxic newts are needed to determine whether the composition and/or structure of the microbiome directly influences newt toxicity.

Additionally, if variation in TTX toxicity is subject to selective forces, TTX-producing symbionts would need to be heritable, directly or indirectly, across generations. The mechanisms underlying microbiome heritability vary from environmental acquisition of microbes across each generation to direct vertical transmission from parent to offspring (*Mandel, 2010*). The development of skin-associated microbial communities in newts, and amphibians more broadly, is not clear, as both host species identity and habitat appear to play important roles across different amphibian taxa (*Ellison et al., 2019*; *Ross et al., 2019*). In newts, one possibility is that TTX-producing bacteria are vertically transferred from females to their eggs, as newt eggs contain TTX and egg toxicity is correlated with the toxicity of the mother (*Gall et al., 2012*; *Hanifin et al., 2003*). Another possibility is that newts possess adaptive traits to facilitate the acquisition and proliferation of TTX-producing bacteria anew from the environment through each generation. Host factors impacting the microbiome may include anti-microbial peptide expression (*SanMiguel and Grice, 2015*) or the production of metabolites that favor TTX-producing microbes. Other traits may influence interspecific interactions within the microbiome to promote colonization and proliferation of TTX-producing symbionts. These traits may be under selective pressure to ultimately benefit TTX-producing symbionts and increase TTX toxicity across newt populations (*Carroll et al., 2003*; *Magarlamov et al., 2017*). Further investigations comparing toxic and non-toxic newts through developmental stages in the wild and in captivity may begin to shed light on this complex process.

Furthermore, because of the challenges of in vitro cultivation and characterization of microbial physiology in symbiotic microbes isolated from their hosts, it is difficult to determine how the dynamics of TTX production are regulated within the in vivo host-associated communities (*Magarlamov et al., 2017*). Under our culture conditions in the lab, we observed TTX production that was typically less than 0.5 ng mL$^{-1}$. However, given that the TTX-producing bacteria identified in this study and in other toxic animals were grown under artificial lab conditions independent of host factors and interactions with other host-associated microbes, estimating the true biosynthetic potential of these TTX-producing bacteria poses a major technical challenge. Identifying the genetic basis of TTX production may help circumvent this problem and allow future researchers to apply sequencing-based metagenomic approaches to determine which organisms are capable of producing TTX (*Chau and Ciufolini, 2011*; *Chau et al., 2011*). These efforts may also facilitate the development of targeted cultivation strategies to better replicate the host environment and more accurately measure TTX production in vitro.

Our results also show that toxic newts possess adaptations in their Na$_v$ channels that confer TTX resistance. The presence of parallel mutations across the Na$_v$ channel family of newts and other TTX-resistant animals suggests that the evolution of resistance involves a highly constrained walk through a narrow adaptive landscape. For example, studies of the skeletal muscle isoform Na$_v$1.4 across a variety of TTX-resistant snake species identify numerous convergent substitutions in the P-loop regions of DIII and DIV, but never in DI or DII (*Feldman et al., 2012*). The Na$_v$1.4 subtype of TTX-resistant newts, including *T. granulosa*, also possess several mutations in DIV and one in DIII, but none in DI or DII. Conversely, mutations in the DI Y/F-371 site are often seen in neural subtypes of TTX-resistant pufferfishes, and we found that this mutation was present in three of the four neural subtypes of newts. Furthermore, when comparing Na$_v$ channel sequences in newts and other TTX-resistant animals, we found that Na$_v$1.6 sequences in newts and garter snakes share two identical substitutions in the P-loops of DIII V1407I and DIV I1699V (*Figure 4—figure supplement 1*). Both newt and snake Na$_v$ sequences were derived from individuals caught in Benton Co., OR, where newts are highly toxic and snakes are highly resistant. These mutations may reflect convergent molecular evolution between predators and prey responding to the same selection pressure. Whether or not these patterns have arisen by chance or through Na$_v$ subtype-specific constraints on P-loop evolution would be interesting to explore in future studies.

Given the potential strength of selection on interactions between newts and their symbiotic microbiota with regard to TTX toxicity, it may be more appropriate to consider the effects of selection across the *hologenome*, the collective genetic variation present in both host and symbionts (*Bordenstein and Theis, 2015*; *Rosenberg and Zilber-Rosenberg, 2013*). Many recent studies emphasize the critical importance of host-associated microbes in basic animal physiology, development, nutrition, nervous system function, and even behavior (*Archie and Theis, 2011*; *Eisthen and Theis, 2016*; *McFall-Ngai et al., 2013*; *Shropshire and Bordenstein, 2016*; *Theis et al., 2016*; *van Opstal and Bordenstein, 2015*). In the coevolutionary arms race between toxic newts and resistant snakes, selection may act upon the phenotype that emerges from the collective interactions between the newt host and bacterial symbionts, termed the *holobiont*. One prediction of the hologenome theory is that adaptive evolution can occur rapidly by increasing the relative abundance of specific symbionts if the metabolites derived from that symbiont are critical for holobiont fitness (*Theis et al., 2016*). This potential evolutionary force would avoid a long and winding road through a complex adaptive landscape for the host, particularly for epistatic traits such as TTX biosynthesis, which is predicted to involve a dozen or more enzymes (*Chau and Ciufolini, 2011*; *Chau et al., 2011*). Future studies exploring the relationship between newt host toxicity and the composition of newt skin microbiota could provide a mechanistic basis for the observed variation in newt toxicity across different populations, revealing potentially interesting cases of parallel evolution occurring at the hologenomic level. Overall, chemical defenses such as neurotoxins provide excellent models for investigating adaptive evolution, as these toxins often target evolutionarily conserved proteins in animal nervous systems, revealing mechanistic associations among protein sequence, physiology, and evolution.

# Materials and methods

**Key resources table**

| Reagent type (species) or resource | Designation | Source or reference | Identifiers | Additional information |
|---|---|---|---|---|
| Strain, strain background (*Escherichia coli*) | STBL2 competent cells | ThermoFisher Scientific | 10268019 | |
| Recombinant DNA reagent | mSCN8A (*Mus musculus*) | DOI: 10.1523/JNEUROSCI.18-16-06093.1998 | | Construct kindly provided by Dr. Al Goldin, UC Irvine |
| Biological sample (*Xenopus laevis*) | Oocytes | xenopus1.com | | |

*Continued on next page*

*Continued*

| Reagent type (species) or resource | Designation | Source or reference | Identifiers | Additional information |
|---|---|---|---|---|
| Sequence-based reagent | 16S_rRNA_8F | Integrated DNA Technologies | 51-01-19-06 | AGAGTTTGATCCTGGCTCAG |
| Sequence-based reagent | 16S_rRNA_515F | DOI: 10.1128/AEM.01043–13 | PCR primer | GTGCCAGCMGCCGCGGTAA |
| Sequence-based reagent | 16S_rRNA_806R | DOI: 10.1128/AEM.01043-13 | PCR primer | TGGACTACHVGGGTWTCTAAT |
| Sequence-based reagent | 16S_rRNA_1492R | Integrated DNA Technologies | 51-01-19-07 | CGGTTACCTTGTTACGACTT |
| Commercial assay or kit | Q5 Site-directed mutagenesis kit | New England Biolabs | E0554S | |
| Commercial assay or kit | T7 mMessage mMachine kit | ThermoFisher Scientific | AM1344 | |
| Chemical compound, drug | Tetrodotoxin | Alomone Labs | T-550 | |
| Software, algorithm | Clampfit v10.7 | Molecular Devices | | |
| Software, algorithm | Geneious v11.0.5 | geneious.com | | |
| Software, algorithm | mothur v1.39.5 | mothur.org | | |
| Software, algorithm | RStudio (v3.6.1) | rstudio.com | | |
| Other | Oasis MCX cartridge | Waters | 186000252 | |
| Other | Acquity UPLC BEH amide column | Waters | 186004801 | |

All procedures involving animals were approved by and conducted under the supervision of the Institutional Animal Care and Use Committee at Michigan State University (approval no. 10/15-154-00), in accordance with guidelines established by the US Public Health Service.

## Laboratory animals

Adult male rough-skinned newts (*Taricha granulosa*) were collected in Oregon, USA (January Pond; 44°36'13.8"N 123°38'12.1"W) under Oregon Department of Fish and Wildlife permit number 104–15. Animals were housed in glass aquaria containing Holtfreter's solution (60 mM NaCl, 0.67 mM KCl, 0.81 mM MgSO$_4$, and 0.68 mM CaCl$_2$; pH 7.2–7.6). Floating platforms in each aquarium provided terrestrial refuges, and newts were maintained at 20°C with a 14:10 light-dark cycle and fed blackworms (*Lumbriculus variegatus*) 2–3 times weekly.

## Cultivation of skin bacteria

To collect bacterial samples, newts were first rinsed in reverse osmosis (RO) H$_2$O for 5 s to remove transient bacteria and swabbed 10 times (down and back) each on the dorsal and ventral skin surfaces using a sterile cotton swab (Puritan Medical Products, Guilford, ME). The sample swab was then placed in 1 mL Hank's Buffered Salt Solution (HBSS; 0.137 M sodium chloride, 5.4 mM potassium chloride, 0.25 mM disodium phosphate, 0.56 M glucose, 0.44 mM monopotassium phosphate, 1.3 mM calcium chloride, 1.0 mM magnesium sulfate, 4.2 mM sodium bicarbonate) and diluted ten-fold over four serial dilutions: $10^{-1}$, $10^{-2}$, $10^{-3}$, and $10^{-4}$. 100 µL of each dilution was then plated on

either R2A agar (0.5 g casein hydrolysate, 0.5 g dextrose, 0.5 g soluble starch, 0.5 g yeast extract, 0.3 g potassium phosphate, 0.3 g sodium pyruvate, 0.25 g casein peptone, 0.25 g meat peptone, 0.024 g magnesium sulfate, 15 g agar, final volume 1 L) or blood agar (10 g peptone, 10 g meat extract, 5 g sodium chloride, 15 g agar, final volume 1 L) infused with defibrinated sheep's blood (10% v/v) (Fisher Scientific, Hampton, NH). Petri dishes containing these mixed community cultures were wrapped in Parafilm to prevent desiccation and incubated at room temperature (20℃) for 1–2 weeks. The combination of nutrient-limited media, cool temperatures, and relatively long incubation periods has been shown to promote microbial diversity and the growth of previously uncultivated microbes (*Sommer, 2015*; *Stevenson et al., 2004*; *Stewart, 2012*).

Following cultivation of mixed communities, individual bacterial colonies were picked and streaked onto new plates to establish pure cultures. Plates were then wrapped in Parafilm and allowed to incubate at 20℃ until colonies appeared. Bacterial stocks were generated by collecting bacterial samples from each streaked plate and submerging in 0.5 mL HBSS with 10% dimethyl sulfoxide (DMSO) for cryoprotection. Samples were then stored at −80℃.

## Taxonomic identification of bacterial isolates

To identify bacterial isolates, we performed colony PCR using the 16S rRNA gene universal primers 8F (5'—AGAGTTTGATCCTGGCTCAG—3') and 1492R (5'—CGGTTACCTTGTTACGACTT—3'). Bacterial colonies were picked with sterile toothpicks and submerged directly into a PCR master mix (final concentration: 1X PCR buffer, 1.5 mM MgCl$_2$, 0.2 mM dNTPs, 0.25 μM forward and reverse primer, 0.05% NP-40, 1.25U Taq polymerase, and nuclease-free H$_2$O). PCR reactions were performed using the following conditions: 3 min at 95℃; 30 s at 95℃, 30 s at 45℃, 1.5 min at 72℃ repeated 30 times; and a final elongation for 5 min at 72℃. PCR products were analyzed by gel electrophoresis and samples yielding products were cleaned using ExoSAP-IT (Affymetrix, Santa Clara, CA) following manufacturer's instructions. DNA samples were submitted to Michigan State University's Genomics Core (East Lansing, MI) for Sanger sequencing using 16S rRNA 8F universal primer (5'—AGAGTTTGATCCTGGCTCAG—3'). Sequences were screened for quality using 4Peaks (Nucleobytes, Amsterdam, Netherlands) and sequences with at least 400 bp of unambiguous base calls after quality trimming were assigned genus-level classifications using the Ribosomal Database Project (RDP) Classifier tool and an 80% confidence threshold (*Cole et al., 2014*).

## Phylogenetic analysis of 16S rRNA gene sequences

Evolutionary relationships among cultured bacteria were inferred by constructing maximum-likelihood phylogenetic trees. Multiple sequence alignments were generated by aligning 16S rRNA gene sequences with the SILVA ribosomal RNA reference database (*Quast et al., 2013*). Gaps and non-informative sites were trimmed to generate the final alignment. Trees were constructed using randomized axelerated maximum-likelihood (RAxML) with 1000 bootstrap replicates (*Stamatakis, 2014*) in Geneious v11.0.5 (*Kearse et al., 2012*) and edited in FigTree v1.4.3 (https://github.com/rambaut/figtree/).

## Sample collection for TTX quantification

To estimate TTX concentrations in newt skin, we followed the non-lethal sampling technique described by Bucciarelli and coworkers (*Bucciarelli et al., 2014*). Animals were first anesthetized in pH-corrected 0.1% tricaine-S (MS-222) dissolved in Holtfreter's solution. Two skin biopsies were then collected from symmetrical sites on the dorsal skin surface, approximately 1 cm laterally from the vertebrae and 1 cm anterior to the hind limbs, using sterile, disposable 2 mm skin biopsy punches (Acu-Punch, Acuderm Inc, Fort Lauderdale, FL). The two skin biopsies from each individual were weighed and then combined in 300 μL 0.1 M acetic acid. Each sample was then placed into a boiling water bath for 5 min followed by an ice bath for an additional 5 min. Subsequent steps were carried out at room temperature. To minimize protein and macromolecular debris, samples were centrifuged at 13,000 x g for 20 min and the supernatant transferred to an Amicon Ultra 10,000 MWCO centrifugal filter (Sigma-Aldrich, St. Louis, MO) followed by a second centrifugation at 13,000 x g for 20 min. Finally, 100 μL 0.1 M acetic acid was added to the filter and a third centrifugation at 13,000 x g for 20 min was performed to wash any remaining TTX. The final sample volume was adjusted to 1 mL before proceeding to solid-phase extraction (below).

To identify TTX-producing bacteria, isolated bacterial strains were revived from frozen stocks and inoculated in 5 ml of R2B broth (0.5 g casein hydrolysate, 0.25 g casein peptone, 0.25 g meat peptone, 0.5 g dextrose, 0.5 g soluble starch, 0.5 g yeast extract, 0.3 g potassium phosphate, 0.3 g sodium pyruvate, 0.024 g magnesium sulfate, final volume 1 L) diluted to either 10% or 50% strength in reverse osmosis (RO) $H_2O$. The use of dilute broth was intended to encourage the production of secondary metabolites. Cultures were grown at room temperature 20°C on a tissue culture rotator for 1 or 2 weeks. After cultivation, each culture was centrifuged at 13,000 x g for 5 min at room temperature, and 1 mL of supernatant was used in solid-phase extraction.

## Solid-phase extraction (SPE)

TTX extractions were performed using a modified solid-phase extraction (SPE) protocol based on that described by *Jen et al. (2008)*. Each skin or bacterial sample was loaded onto a mixed cation exchange cartridge (Oasis MCX cartridges, Waters, MA) previously regenerated with 1 mL of methanol and equilibrated with 1 mL RO $H_2O$. Samples were drawn through the cartridge over 30 s using a Vac-Man laboratory vacuum manifold (Promega, Madison, WI) coupled with VacConnectors (Qiagen, Germantown, MD). Each cartridge was then washed with 1 mL acetonitrile, 1 mL methanol, and 1 mL distilled $H_2O$. TTX was eluted twice from the cartridge with 0.125 mL 0.2 M HCl in 20% methanol. Both eluates were combined and dried in a SpeedVac vacuum centrifuge (Savant SpeedVac SC110, Thermo Fisher Scientific, Waltham, MA), then resuspended in 0.2 mL 0.5% acetic acid in water. 50 μL aliquots of each sample were prepared for LC-MS/MS analysis.

## Liquid chromatography tandem mass spectrometry (LC-MS/MS)

TTX analyses were performed using a Waters TQ-D mass spectrometer coupled to a Waters ACQUITY UPLC system with a binary solvent manager. Chromatographic separations were performed on a Waters ACQUITY UPLC BEH amide column (2.1 × 100 mm; 1.7 μm particles; Waters Co., Milford, MA); column temperature was held at 40°C. For liquid chromatography, we used 0.1% formic acid in water (mobile phase A) and acetonitrile (mobile phase B) with a flow rate of 0.4 mL/min. The injection volume was set to 10 μL. The linear gradient elution program was as follows (A/B): 0–1.0 min (5/95), 1.0–1.5 min (50/50), 1.5–2.0 min (55/45), 2.0–3.5 min (60/40), 3.5–4.0 min (65/35) before the gradient returned to the initial condition (5/95). TTX was analyzed in positive electrospray ionization mode using multiple reaction monitoring with a transition of 320.1 > 162.1 (cone voltage: 50 eV; collision energy: 40 eV) as the primary channel for quantification and 320.1 > 302.1 (cone voltage: 50 eV; collision energy: 40 eV) as the secondary channel for confirmation. The capillary voltage was 3.0 kV. Source and desolvation temperatures were 130°C and 500°C, respectively; cone gas and desolvation gas flows were 40 and 700 L/hr, respectively. Data were acquired using MassLynx 4.1 software (Waters Co.). Extracts from bacterial and skin samples were compared with TTX analytical standards acquired from Sigma-Aldrich (St. Louis, MO). A calibration curve was included in each LC-MS/MS run with the following concentrations: 0.01, 0.05, 0.1, 0.5, 1, 2.5, 5, 10, and 25 ng/ml. Concentrations of TTX quantified from skin biopsies were normalized relative to tissue mass. The presence of TTX in skin samples and bacterial cultures was confirmed by a retention time identical to that of authentic TTX as well as the presence of both primary and secondary ion transitions. All chromatograms were plotted in R v3.4.1.

## Scanning electron microscopy

3 × 3 mm skin samples were dissected from the dorsal region of a euthanized newt. Each sample was fixed in 4% glutaraldehyde in 0.1 M sodium phosphate buffer (pH 7.4) overnight at 4°C. Following fixation, samples were briefly rinsed in 0.1 M sodium phosphate buffer and dehydrated in an ethanol gradient (25, 50, 75, 95, 100, 100, 100%) for 10 min each. Any remaining liquid in the samples was removed by critical point drying in a Balzers Model 010 critical point dryer (Balzers Union Ltd., Balzers, Liechtenstein) using carbon dioxide as the transitional fluid. Each skin sample was then mounted on an aluminum stub using carbon suspension cement (SPI Supplies, West Chester, PA) and coated with platinum (8 nm thickness) using a Q150T turbo pumped sputter coater (Quorum Technologies, Laughton, East Sussex, England) purged with argon gas. Samples were examined and images obtained using a JEOL JSM-7500F cold field emission scanning electron microscope (JEOL Ltd, Tokyo, Japan).

## Microbiome sample collection

Skin bacterial samples were collected from two populations of rough-skinned newts, one in Oregon (January Pond; 44°36'13.8"N 123°38'12.1"W) and one in Idaho (Virgil Phillips Farm Park, Idaho; 46° 48'49.9"N 117°00'57.2"W) under Oregon Department of Fish and Wildlife permit number 104–15 and Idaho Department of Fish and Game Wildlife Bureau permit number 150521, respectively. Microbial samples were collected from January Pond, Oregon in Summer 2013 and Virgil Philips Farm Park, Idaho in Fall 2016. Animals were caught in ponds with dipnets or minnow traps, and each animal was handled with a fresh pair of nitrile gloves. Bacterial samples were collected from two skin sites (dorsal and ventral) and from the surfaces of two external glands (submandibular gland and cloaca) for a total of four samples per animal. Sterile cotton-tipped swabs were dipped into fresh aliquots of filter-sterilized wetting solution (0.15M NaCl and 0.1% Tween-20) and stroked across each body surface 20 times. Each swab was then placed into a sterile 1.5 mL conical tube and kept on dry ice until transported to the lab, where they were stored at −80°C. In addition to swabs from newts, we also collected soil samples from pond sediment and pond water samples from each site in sterile 50 mL conical tubes.

## Bacterial DNA extraction

Total DNA from swab samples was extracted using a QIAamp DNA Mini Kit (Qiagen) as follows. First, 500 µl TE buffer (10 mM Tris-HCl, 50 mM EDTA, pH 8, 0.2 µm filter-sterilized) was added to each cotton swab sample and pulse vortexed for 15 s. The buffer was then transferred to a sterile bead-beating tube containing 750 mg zirconia silica beads (0.1 mm, BioSpec, Bartlesville, OK) and each sample underwent bead-beating for 60 s on a Thermo Savant FastPrep FP120 (Thermo Fisher, Waltham, MA) at setting 5. Samples were briefly centrifuged and the lysate transferred to a new 2 mL tube. 25 µL proteinase K and 500 µL kit buffer AL were added to each sample, and samples were then pulse vortexed for 15 s and incubated at 56°C for 10 min on a heat block. Each lysate was then acidified by adding 100 µL sodium acetate (3M, pH 5.5), followed by 500 µL 100% ethanol. Samples were pulse vortexed for 15 s and applied to QIAamp mini spin columns attached to a vacuum manifold via a sterile VacConnector (Qiagen) to a Luer valve. The entire lysate was pulled through the column by application of a vacuum and then each column was washed with 750 µL Buffer AW1 and Buffer AW2, respectively. Next, the spin column was transferred to a clean collection tube and centrifuged at 6000 x *g* for 1 min in a bench-top microcentrifuge to dry the membrane. After drying, the spin column was placed into a clean 1.5 mL microcentrifuge tube, 50 µL nuclease-free $H_2O$ was applied to the membrane, and the column was incubated for 5 min at 20°C. Each tube was then centrifuged at 10,000 rpm for 1 min to elute the DNA. For soil and water samples, DNA extraction was performed using the MoBio DNeasy PowerSoil Kit (Qiagen) per manufacturer's instruction. For soil samples, 0.2 g of pond sediment was directly added to the PowerBead tubes provided by the kit; for pond water, we centrifuged 15 mL pond water at 10,000 x g for 10 min at 4°C and resuspended the bacterial cell pellet in 500 µl TE buffer, which was then transferred to a bead beating tube. For negative controls, we performed DNA extractions and PCR reactions on cotton swab samples prepared in the field. We included PCR products from these negative controls with each batch of bacterial samples and included the resulting products in our 16S rRNA gene amplicon library preparation and sequencing.

## 16S rRNA amplicon library preparation

Illumina paired-end reads overlap in the V4 region, allowing for poor quality base calls to be discarded in favor of higher quality sequence on the opposite strand. A dual-barcoded two-step PCR was therefore conducted to amplify the V4 hypervariable regions of the bacterial 16S rRNA gene. V4 primers were designed based on those provided by *Kozich et al. (2013)* with the addition of adapter sequences for dual-index barcodes (515F–ACACTGACGACATGGTTCTACAGTGCCAGC MGCCGCGGTAA; 806R–TACGGTAGCAGAGACTTGGTCTTGGACTACHVGGGTWTCTAAT). In a dedicated PCR hood, 2 µl DNA extract was added to a PCR mixture containing 0.05 µM primers (Integrated DNA Technologies, Coralville, IA), and Q5 Hot Start High Fidelity 2X Master Mix (New England Biolabs, Ipswich, MA) diluted with nuclease-free water to a 1X final concentration (25 µl final volume). PCR was conducted using a Veriti thermal cycler (Applied Biosystems, Foster City, CA) under the following conditions: 98°C for 30 s; then 98°C for 10 s, 51°C for 20 s, and 72°C for 20 s for

15 cycles. The machine was then paused and 2 µl primers (2 µM) with dual-index barcodes and Illumina sequencing adapters (University of Idaho IBEST Genomics Resources Core Facility) were added to each reaction, bringing the final reaction volume to 25 µl. Amplification resumed with 98°C for 30 s; then 98°C for 10 s, 60°C for 20 s, and 72°C for 20 s for 15 cycles; then a final extension step of 72°C for 2 min. Samples were held at 4°C in the thermocycler until being stored at −20°C. Quality of PCR amplicons was evaluated using a QIAxcel DNA screening cartridge (Qiagen) and DNA quantified using a Qubit fluorometer (Invitrogen, Carlsbad, CA) and the Qubit dsDNA High Sensitivity Assay (Thermo Fisher Scientific, Waltham, MA).

Equimolar volumes of each PCR product containing 50 ng DNA were pooled to create a composite sample for high-throughput sequencing and submitted to the University of Idaho IBEST genomics core. Amplicon pools were size-selected using AMPure beads (Beckman Coulter, Brea, CA). The cleaned amplicon pool was quantified using the KAPA Illumina library quantification kit (KAPA Biosciences, Roche, Basel, Switzerland) and StepOne Plus real-time PCR system (Applied Biosystems, Foster City, CA).

## 16S rRNA gene Illumina sequencing

Sequences were obtained using an Illumina MiSeq (San Diego, CA) v3 paired-end 300 bp protocol for 600 cycles. Raw DNA sequence reads were processed using the Python application dbcAmplicons (https://github.com/msettles/dbcAmplicons), which was designed to process Illumina double-barcoded amplicons generated in the manner described above. For sequence pre-processing, barcodes were allowed to have at most 1 mismatch (Hamming distance) and primers were allowed to have at most 4 mismatches (Levenshtein distance) as long as the final 4 bases of the primer perfectly matched the target sequence. Reads lacking a corresponding barcode and primer sequence were discarded. V4 sequences were processed in mothur (v 1.39.5) following the MiSeq protocol (*Kozich et al., 2013*). Paired sequence reads were joined into contigs, screened for quality and removal of chimeras, then aligned to the SILVA 16S ribosomal RNA database (*Quast et al., 2013*) and clustered into operational taxonomic units (OTUs) based on 97% nucleotide identity. Taxonomic assignment of OTUs was then performed using the RDP classifier (*Cole et al., 2014*).

## Statistical analysis of newt-associated bacterial communities

Prior to analysis, each 16S rRNA gene amplicon profile was subsampled to 5000 sequences. Rarefaction and Good's coverage analyses were conducted using the rarefaction.single() and summary.single() commands in mothur, respectively. Relative abundances of bacterial OTUs were calculated and visualized using the Phyloseq package in R (v3.4.1) (*McMurdie and Holmes, 2013*). All subsequent microbial ecology analyses were all conducted in R using the vegan package (v2.5–3) (*Oksanen et al., 2014*), and plots were produced using the ggplot package (*Wickham and Chang, 2007*). Beta diversity matrices were produced using Jaccard and Bray-Curtis dissimilarity indices (*Whittaker, 1972*). Principal coordinates analyses (PCoA) were conducted on each dissimilarity matrix, and significant differences between groups were determined using a permutational multivariate analysis of variance (PERMANOVA) with 9999 permutations and a $p < 0.05$ cutoff. A permutation test for multivariate dispersion (PERMDISP) was conducted to test for differences in variance among community samples. Linear discriminant analysis effect size (LEfSe) was performed on the Galaxy server (http://huttenhower.sph.harvard.edu/galaxy/).

## RNA extraction from newt tissue

Newts were euthanized by immersion in pH-corrected 0.1% MS-222, and tissue samples including brain, nose, heart, and skeletal muscle were collected and stored in RNAlater (ThermoFisher Scientific, Waltham, MA) at −20 °C. Total RNA was extracted from newt tissues using TRIzol (Thermo-Fisher Scientific). Briefly, each tissue was aseptically dissected and placed into a sterile tube containing 1 mL TRIzol reagent. Tissues were homogenized in TRIzol using a TissueRuptor (Qiagen, Hilden, Germany), and total RNA was extracted following manufacturer's instructions. The clean RNA pellet was resuspended in 50 µL tris-EDTA buffer (pH 8.0) and total RNA yield was quantified by fluorescence using a Qubit fluorometer (ThermoFisher Scientific).

## Brain and nose transcriptome sequencing

We generated reference transcriptomes from the brain and nose of *T. granulosa* for identification of *SCN* genes. Non-excitable tissues including liver and skin were included in the sequencing run but were not used for analysis in this study. Poly-adenylated RNA was purified from total RNA samples (previous section) using the NEXTflex PolyA Bead kit (Bioo Scientific, Austin, TX) according to the manufacturer's instructions. Lack of contaminating ribosomal RNA was confirmed using the Agilent 2100 Bioanalyzer. Strand-specific libraries for each sample were prepared using the dUTP NEXTflex RNAseq Directional kit (Bioo Scientific), which includes magnetic bead-based size selection, resulting in an average library size of 462 bp. Libraries were pooled in equimolar amounts after quantification using the fluorometric Qubit dsDNA high sensitivity assay kit (Life Technologies) according to the manufacturer's instructions. Libraries were sequenced on an Illumina HiSeq 2000 (Harvard University, Cambridge, MA) in one lane to obtain 503,241,123 paired-end 100 bp reads.

## Transcriptome assembly and annotation

We first corrected errors in the Illumina reads using Rcorrector (*Song and Florea, 2015*); parameters: run_rcorrector.pl -k 31) and then applied quality and adaptor trimming using Trim Galore! (http://www.bioinformatics.babraham.ac.uk/projects/trim_galore/; parameters: `trim_galore – paired –phred33 –length 36 -q 5 –stringency 5 –illumina -e 0.1`). After filtering and trimming, a total of 500,631,191 reads remained for de novo assembly. We created the newt transcriptome de novo assembly using Trinity, (`parameters: –seqType fq –SS_lib_type RF`). The raw Trinity assembly produced 2,559,666 contigs (N50: 399 bp). To reduce redundancy in the assembly, we ran cd-hit-est (50) (parameters: -c 0.97), resulting in 2,208,791 contigs (N50: 390 bp). As the sequences were obtained from a wild-caught newt, we next filtered the assembly to remove parasites, microbes, and other contaminants. To accomplish this, we used BLAST to compare each contig with proteins in the Uniprot SwissProt database (e-value threshold of 1e-5); specifically, we used non-vertebrate reference genomes, including those of arthropods (*Drosophila*), microbes (fungi, *Saccharomyces*; bacteria, *Pseudomonas*), and parasites (*Caenorhabditis*) to identify potential contaminants, resulting in the removal of 61,185 contigs. For the purposes of our study, we only retained contigs with homologs to vertebrate proteins based on this BLAST search of the Swiss Prot database; our final draft assembly of the newt transcriptome contained 77,535 contigs with an N50 of 3025 bp. We assessed the completeness of this final assembly by examining vertebrate ortholog representation using BUSCO (*Simão et al., 2015*), which showed 86% of BUSCO groups represented in the assembly.

## PCR and cloning of *SCN* genes

To evaluate *SCN* gene sequence assembly and confirm the presence of P-loop mutations, we used sequence-specific PCR primers to amplify the DI-DIV transmembrane sequences of each gene from newt cDNA, followed by Sanger sequencing. cDNA templates were synthesized from newt brain and nose RNA using the SuperScript III First-Strand Synthesis kit following manufacturer's instructions (Invitrogen, Carlsbad, CA). RNA (0.5–1 μg) was primed with oligo(dT)$_{20}$ primers, targeting the mRNA poly(A)+ tail to enhance synthesis of expressed mRNA transcripts. cDNA samples were stored at −20 °C until use. PCR reactions were performed using Q5 High-Fidelity 2X master mix (New England Biolabs, Ipswich, MA) and analyzed by gel electrophoresis on 0.8% w/v agarose gel in tris-acetate-EDTA buffer (pH 8.0). Amplified DNA was either sequenced directly from PCR products or cloned into the pGEM-T DNA vector (Promega, Madison, WI). In the latter case, PCR products were first purified by spin column using the DNA Clean and Concentrator kit (Zymo Research, Irvine, CA), then A-tailed using GoTaq (Promega) by combining 10 μL purified PCR product, 2.5 μL 10X buffer, 5 μL dATP (1 mM), 0.2 μL Taq polymerase, and 7.3 μL nuclease-free water to a total volume of 25 μL, and then incubated at 72 °C for 20 min. A-tailed products were used for TA cloning using the pGEM Easy Vector system (Promega). Ligated PCR products were transformed into STBL2 competent *E. coli* cells by heat shock at 42 °C for 45 s, and 950 μL of SOC media was added to each sample. The following procedures were then adjusted specifically for *SCN* gene cloning based on published recommendations (*Feldman and Lossin, 2014*): samples were incubated at 30 °C for 60 min and plated on ½ strength antibiotic Luria-Bertani (LB) agar plates (50 μg/mL ampicillin or 7.5 μg/mL tetracycline). Plates were incubated at 30 °C for two days, and smaller colonies were

preferentially selected over large colonies. Plasmid DNA was recovered using the QIAprep Spin Miniprep kit (Qiagen) and quantified using a Qubit fluorometer. Aliquots of transformed competent cells in LB were combined with equal volumes of 50% glycerol and stored at −80 ℃. PCR products or cloned PCR amplicons were submitted for Sanger Sequencing at the Michigan State University Genomics Core Facility (East Lansing, MI).

### *SCN* sequence analysis

Sanger sequences from *SCN* gene PCR products were analyzed in Geneious v11.0.5 (Biomatters Inc, Newark, NJ). We identified full length coding sequences for all genes except *SCN2A* (encodes all four transmembrane domains and P-loop regions) and *SCN1A* (only encodes DIII and DIV of the channel). The sequence for *T. granulosa* $Na_v1.4$ was obtained from GenBank (KP118969.1) for comparison with other newt channels. All other GenBank accession numbers for vertebrate $Na_v$ channel sequences are shown in *Figure 4—source data 1*. Consensus sequences for each *T. granulosa SCN* gene are available on GenBank (accession numbers MT125668-72).

We assessed the quality of sequence base calls by peak shape in the sequence electropherogram files. To identify mutations in newt $Na_v$ channels, sequences were translated and aligned with orthologous $Na_v$ sequences from representative vertebrate taxa: human (*Homo sapiens*), mouse (*Mus musculus*), chicken (*Gallus gallus*), green anole (*Anolis carolinensis*), and the two amphibians for which complete *SCN* gene sequence data were available in GenBank, the Western clawed frog (*Xenopus tropicalis*) and Tibetan plateau frog (*Nanorana parkeri*). We aligned all $Na_v$ protein sequences using MUSCLE (v3.8.425) and extracted the P-loop regions for analysis of *T. granulosa* mutations. Mutations were annotated and numbered by reference to the homologous amino acid site in the mouse $Na_v1.6$ channel (GenBank accession: U26707.1).

### Site-directed mutagenesis

Characterization of the effects of three individual mutations in *T. granulosa* $Na_v1.6$ on TTX binding were examined by heterologous expression and electrophysiological recording in *Xenopus laevis* oocytes. We introduced these mutations into an orthologous *M. musculus SCN8A* construct (*mSCN8A*), kindly provided by Dr. Al Goldin (*Smith et al., 1998*), to create a chimeric newt-mouse *SCN8A* construct. The *mSCN8A* construct contained an upstream T7 promotor and a downstream NotI restriction enzyme site for plasmid linearization. Site-directed mutagenesis (SDM) was performed using the Q5 SDM kit (New England Biolabs). SDM primers containing the target mutation were produced using the NEBase Changer tool (https://nebasechanger.neb.com). After PCR amplification, PCR products were treated with kinase-ligase-DpnI enzyme mix (New England Biolabs), then purified by spin column using the DNA Clean and Concentrator kit (Zymo Research). Mutated plasmid DNA was cloned into STBL2 *E. coli* competent cells by heat shock at 42 ℃ for 45 secs. Incubation and colony selection was performed following the protocol of *Feldman and Lossin (2014)*; specifically, all incubations were performed at 30 ℃ using ½ antibiotic (ampicillin: 50 mg $L^{-1}$; tetracycline: 5 mg $L^{-1}$) and two-day incubation periods. Colonies were picked and submerged in LB for overnight incubation, and plasmid DNA was recovered by mini-prep (Qiagen). Samples of each culture were combined with an equal volume of 50% glycerol and stored at −80 ℃.

Because rearrangements and other replication errors are common with sodium channel sequences (52), each plasmid was screened by restriction enzyme (RE) digest using BamH1 and IgIII (New England Biolabs) and run on a 0.8% w/v agarose gel to ensure the correct fragmentation pattern was present. Samples with the correct RE pattern were inoculated in 400 mL of LB and incubated overnight, and plasmid DNA was recovered using the Qiagen plasmid maxi-prep kit (Qiagen). Maxi-prepped DNA was quantified using a Qubit fluorometer and the *mSCN8A* reading frame was sequenced at the MSU Genomics Core Facility to ensure the correct substitution was made and that no other mutations were introduced into the construct.

### cRNA synthesis

Capped mRNA (cRNA) was synthesized from linearized DNA templates. Plasmid DNA containing the unmutated *mSCN8A*, individual mutations, or the triple-mutant construct was linearized by overnight digestion using the NotI restriction enzyme (New England Biolabs). 10% SDS and proteinase K were added to each reaction and incubated at 50 ℃ for 1 hr. Two volumes of phenol were added

and mixed into each sample prior to centrifugation at 12,000 x *g* at 4 °C for 10 min, and the upper aqueous layer was transferred to a new tube. Linearized DNA was precipitated by the addition of two volumes of ice cold 100% ethanol, 20 μL of 3M sodium acetate, and 1 μL of glycogen, followed by overnight incubation at −20 °C. Samples were then centrifuged at 12,000 x *g* at 4 °C for 20 min, the supernatant discarded, and the DNA pellet washed with 0.5 mL 75% ethanol by brief vortexing and re-centrifugation. The supernatant was discarded, and the DNA pellet air-dried and resuspended in 10 μL nuclease-free water. cRNA was produced using the T7 mMessage mMachine kit following manufacturer's instructions (ThermoFisher Scientific). Reaction components were combined with 250 ng of linearized template DNA and incubated at 37 °C for 2–8 hr. Synthesized RNA was recovered by lithium chloride precipitation with the addition of 1 μL glycogen. RNA pellets were resuspended in 20 μL nuclease-free water and aliquots at 50 ng/μL concentration were produced for injection into oocytes. Aliquots were stored at −80 °C.

## Na$_v$ channel expression in *Xenopus* oocytes

Ovaries of adult *Xenopus laevis* were purchased from *Xenopus* 1 (Dexter, MI) for electrophysiological recordings. Individual oocytes were collected from the ovary by enzymatic digestion using collagenase (0.4 mg mL$^{-1}$, type II activity 255 μ/mg) in Ca$^{2+}$-free ND96 solution (in mM: 96 NaCl, 2 KCl, 1.8 CaCl2, 1 MgCl2, and 5 HEPES adjusted to pH 7.5 and supplemented with 0.1 mg/ml gentamycin, 0.55 mg/ml pyruvate, and 0.5 mM theophylline). After 90 min incubation, treated ovaries were washed 5X with normal ND96, and Stage 5 and 6 oocytes were selected for injection.

Oocytes were injected with cRNA samples using a Nanoject III (Drummond Scientific, Broomall, PA). Nanoject glass capillaries (Drummond, 3-000-203-G/X) were pulled into pipettes using a Sutter P-97 micropipette puller (Sutter Instruments Co., Novato, CA) using the following conditions: Heat = ramp+5, Pull = 100, Velocity = 50, Delay = 50, Pressure = 500. Pipettes were backfilled with mineral oil and placed onto the Nanoject. A 4 μL droplet of each cRNA sample (50 ng/μL) was front loaded into the pipette, and oocytes were injected with either 10, 25, or 50 nL of cRNA (0.5–2.5 ng/oocyte). Oocytes were incubated at 14 °C in ND96 and used for recording within 2–10 days.

## Electrophysiology

Macroscopic sodium currents were measured by two-electrode voltage clamp using a Warner Instruments Oocyte Clamp (model OC-725C). Borosilicate glass pipettes (1B120F-4, World Precision Instruments, Sarasota, FL) pulled to a 1 or 2 MΩ tip (Heat = Ramp + 5, Pull = 100, Vel = 50, Time = 50, on a Sutter Instruments P-97 puller) served as current and voltage electrodes, respectively. Pipettes were filled with 3M KCl and 0.5% agarose. Oocytes were recorded in a RC-26Z diamond bath recording chamber with a chamber volume of 350 μL (Warner Instruments, Hamden, CT) in filter-sterilized ND96 recording solution (96 mM NaCL, 2 mM KCl, 1.8 mM CaCal2, 1 mM MgCl2, and 10 mM HEPES; pH 7.5) at room temperature (20°−22 °C). Purified TTX (Sigma Aldrich or Abcam, Cambridge, UK) was diluted in recording solution and perfused through the chamber for experimental applications. Na$^+$ current traces were digitized at 10 kHz using a Digidata 1550B (Molecular Devices, San Jose, CA) and recorded in pCLAMP v10.7 (Molecular Devices). Leak currents were subtracted by P/4 correction.

The electrophysiological properties measured from each construct (wildtype, triple mutant, and three individual mutants of *mSCN8A*) include peak Na$^+$ current (I$_{max}$), conductance, and the voltage-dependence of fast inactivation. Current-voltage (I/V) relationships for each *mSCN8A* construct were determined using an activation protocol in which each oocyte was clamped to a membrane potential of −100 mV and depolarized from −80 to +65 mV in 5 mV steps. The pulse duration was 50 ms with an inter-pulse interval of 5 s. Fast inactivation was measured by clamping the membrane at increasingly depolarized membrane potentials from −100 mV to +10 mV in 5 mV steps for 100 ms followed by a 50 ms test pulse at 0 mV. In such a protocol, the current generated during the test pulse is inversely related to the proportion of channels that are inactivated.

To measure the effects of TTX on I$_{max}$, 10 chamber volumes (3.5 mL) of ND96 recording solution containing experimental concentrations of TTX (100 nM, 1 μM, and 10 μM) were perfused over the oocyte at a flow rate of 5 mL per min. TTX block was monitored by delivering a 50 ms test pulse of 0 mV from a holding potential of −100 mV every 10 secs. Currents were considered to have reached steady-state block when we observed no change in peak current for 10 consecutive pulses,

approximately 5 mins after the application of TTX. Step activation and fast inactivation were then measured in the presence of TTX using the protocols described above. To determine whether TTX was responsible for reductions in $I_{max}$, all oocytes were then washed for 5–10 mins with normal ND96 and re-recorded by the same protocols. Recordings were discontinued if the membrane leak potential increased more than 0.1 mV during the recording.

### Electrophysiology data analysis

Data were extracted from $Na^+$ current traces in Clampfit v10.7 (Molecular Devices) and exported for analysis in R Studio v3.6.0. Peak $Na^+$ currents elicited by each voltage step in the presence of TTX were normalized relative to the maximum peak current ($I_{max}$) recorded for each oocyte prior to the addition of TTX. Statistical differences in peak current in the presence and absence of TTX were determined using one-way repeated measures analysis of variance (ANOVA) at the $-20$ mV depolarization step, at which $I_{max}$ was typically largest, followed by a post-hoc Tukey's test with Bonferroni correction.

Normalized conductance curves for each oocyte were determined by $G_{Na} = I_{max}/(V-V_{Na})$, where $I_{max}$ is the peak current, V is the voltage step, and $V_{Na}$ is the $Na^+$ reversal potential. $V_{Na}$ was assessed empirically for each oocyte from the corresponding I/V curve. Conductance-voltage plots were fit with a single Boltzman equation, $G_{Na} = 1/(1 + exp[-(V – V_{1/2})/k])$, where V is the voltage step, $V_{1/2}$ is the voltage required for half-maximal activation, and k is the slope of the Boltzmann fit. The $IC_{50}$ for TTX binding was determined from the ratio of peak currents in the presence and absence of TTX by a single-site Langmuir equation, $IC_{50} = [TTX](I_{TTX}/I_{max}) / (1 – (I_{TTX}/I_{max}))$, where $I_{max}$ is the peak current recorded under control conditions and $I_{TTX}$ is the current recorded for a given concentration of TTX. The ratio of $I_{TTX}/I_{max}$ across concentrations of 0.1, 1, 10, and 30 µM were fit with a single Hill equation to generate $IC_{50}$ values using the *drc* package in R (*Ritz et al., 2015*).

## Acknowledgements

We thank Drs. Carol Flegler, A Daniel Jones, and Chen Zhang for technical assistance, Dr. Emma Coddington for providing newts and for assistance in sample collection, Yannik Roell for assistance in bacterial sample collection in the field, Dr. Mark McGuire for the use of his laboratory to process bacterial DNA samples, and Dr. James Kremer for helpful discussions. We also thank the undergraduate students who have contributed to the newt bacterial culture collection, particularly Alyssa Garvey, Kimberly Brummell, Jennifer Lough, and Ayley Shortridge, who also prepared the illustration in Figure 3A. We thank Dr. Alan Goldin (University of California, Irvine) for providing constructs for the mouse SCN8A α subunit and the rat β1 and β2 subunits, and Drs. Ke Dong and Ashlee Rowe for their generosity in sharing reagents and protocols. We are indebted to Dr. Abhijna Parigi for her assistance with electrophysiological recordings. We also thank Drs. Harold Zakon and Ben Liebeskind (University of Texas) for generating a preliminary transcriptome of the *T. granulosa* nose. Finally, we thank Dr. Nicholas Bellono (Harvard University) for helpful comments on the manuscript. This work was supported by the BEACON Center for the Study of Evolution in Action under US National Science Foundation Cooperative Agreement No. DBI-0939454; National Science Foundation grants IOS-1354089 and IOS-1655392 to HLE and IOS-0920505 to KRT; a Bauer Fellowship from Harvard University to LAO; and an NSF Graduate Research Fellowship to PMV (Fellow ID: 2014165835).

## Additional information

### Competing interests

Lauren A O'Connell: Reviewing editor, *eLife*. The other authors declare that no competing interests exist.

## Funding

| Funder | Grant reference number | Author |
| --- | --- | --- |
| National Science Foundation | DBI-0939454 | Patric M Vaelli<br>Kevin R Theis<br>Heather L Eisthen |
| National Science Foundation | IOS-1354089 | Heather L Eisthen |
| National Science Foundation | IOS-1655392 | Heather L Eisthen |
| National Science Foundation | IOS-0920505 | Kevin R Theis |
| Harvard University | Bauer Fellowship | Lauren A O'Connell |
| National Science Foundation | Graduate research fellowship, fellow ID: 2014165835 | Patric M Vaelli |

The funders had no role in study design, data collection and interpretation, or the decision to submit the work for publication.

## Author contributions

Patric M Vaelli, Conceptualization, Data curation, Formal analysis, Funding acquisition, Investigation, Methodology, Writing - original draft, Writing - review and editing; Kevin R Theis, Conceptualization, Supervision, Funding acquisition, Writing - review and editing; Janet E Williams, Formal analysis, Supervision, Investigation, Writing - review and editing; Lauren A O'Connell, Formal analysis, Funding acquisition, Investigation, Writing - review and editing; James A Foster, Conceptualization, Funding acquisition, Writing - review and editing; Heather L Eisthen, Conceptualization, Resources, Supervision, Funding acquisition, Writing - original draft, Project administration, Writing - review and editing

## Author ORCIDs

Patric M Vaelli (iD) https://orcid.org/0000-0002-3837-4564
Lauren A O'Connell (iD) https://orcid.org/0000-0002-2706-4077
Heather L Eisthen (iD) https://orcid.org/0000-0002-9049-5750

## Ethics

Animal experimentation: All procedures involving animals were approved by and conducted under the supervision of the Institutional Animal Care and Use Committee at Michigan State University (approval no. 10/15-154-00), in accordance with guidelines established by the US Public Health Service. Newts were collected under Oregon Department of Fish and Wildlife permit number 104-15 and Idaho Department of Fish and Game Wildlife Bureau permit number 150521.

## Decision letter and Author response

Decision letter https://doi.org/10.7554/eLife.53898.sa1
Author response https://doi.org/10.7554/eLife.53898.sa2

# Additional files

## Supplementary files

• Transparent reporting form

## Data availability

Sequences of newt voltage-gated sodium channels have been submitted to NCBI GenBank and are available under the accession numbers MT125668 - MT125672. Bacterial OTU abundance tables and accompanying taxonomic information have been uploaded to Dryad under the DOI: https://doi.org/10.5061/dryad.pg4f4qrk1.

The following datasets were generated:

| Author(s) | Year | Dataset title | Dataset URL | Database and Identifier |
|---|---|---|---|---|
| Vaelli PM, Theis KR, Williams JE, O'Connell LA, Foster JA, Eisthen HL | 2020 | The skin microbiome facilitates adaptive tetrodotoxin production in poisonous newts | https://doi.org/10.5061/dryad.pg4f4qrk1 | Dryad Digital Repository, 10.5061/dryad.pg4f4qrk1 |
| Vaelli PM, Theis KR, Williams JE, O'Connell LA, Foster JA, Eisthen HL | 2020 | Taricha granulosa voltage-gated sodium channel alpha subunit SCN1A mRNA, partial cds | https://www.ncbi.nlm.nih.gov/nuccore/MT125668 | NCBI GenBank, MT125668 |
| Vaelli PM, Theis KR, Williams JE, O'Connell LA, Foster JA, Eisthen HL | 2020 | Taricha granulosa voltage-gated sodium channel alpha subunit SCN2A mRNA, partial cds | https://www.ncbi.nlm.nih.gov/nuccore/MT125669 | NCBI GenBank, MT125669 |
| Vaelli PM, Theis KR, Williams JE, O'Connell LA, Foster JA, Eisthen HL | 2020 | Taricha granulosa voltage-gated sodium channel alpha subunit SCN3A mRNA, complete cds | https://www.ncbi.nlm.nih.gov/nuccore/MT125670 | NCBI GenBank, MT125670 |
| Vaelli PM, Theis KR, Williams JE, O'Connell LA, Foster JA, Eisthen HL | 2020 | Taricha granulosa voltage-gated sodium channel alpha subunit SCN5A mRNA, complete cds | https://www.ncbi.nlm.nih.gov/nuccore/MT125671 | NCBI GenBank, MT125671 |
| Vaelli PM, Theis KR, Williams JE, O'Connell LA, Foster JA, Eisthen HL | 2020 | Taricha granulosa voltage-gated sodium channel alpha subunit SCN8A mRNA, complete cds | https://www.ncbi.nlm.nih.gov/nuccore/MT125672 | NCBI GenBank, MT125672 |

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
