## [Decision Letter]

**Acceptance summary:**

This work sheds light on the importance of the skin microbiome in the defense of newts against predators. It elegantly shows that microorganisms may be crucial drivers of predator-prey interactions and complement our current models of trophic interactions.

**Decision letter after peer review:**

Thank you for submitting your article "The skin microbiome facilitates adaptive tetrodotoxin production in poisonous newts" for consideration by *eLife*. Your article has been reviewed by two peer reviewers, including Christelle Robert as the Reviewing Editor and Reviewer #1, and the evaluation has been overseen by Ian Baldwin as the Senior Editor. The following individual involved in review of your submission has agreed to reveal their identity: Mabel Cristina GonzÃ¡lez Montoya (Reviewer #2).

The reviewers have discussed the reviews with one another and the Reviewing Editor has drafted this decision to help you prepare a revised submission.

Summary:

The manuscript of Vaelli and colleagues takes an important step in understanding the production of TTX in amphibians. The authors characterized the skin microbiome of a toxic and of a non-toxic newt population. They identified TTX producing bacteria from 4 distinct genera from the skin surface of toxic newts. They further investigated and identified new mutations of the NA_v_ channels underlying auto-resistance to TTX.

The manuscript is really exciting and represents an important contribution to the field. Yet, some concerns should be addressed prior to publication.

Essential revisions:

1) The comparison between one toxic and one non toxic population of newts collected at different locations, but most importantly, at different time points (3 years apart and over different seasons) weakens the manuscript. Is there any evidence for the skin microbiome to be stable over such time/seasons that can be used to support this comparison? We believe that the manuscript is strong enough when focusing on toxic newts. We suggest the authors to move the data obtained about non toxic newts (half of Figure 1H and Figure 2) to the supplementary material. Please mention in the legend of these supplementary data that the 2 newt populations were collected at different time points.

2) The comparison is also confusing as one can't extract the information regarding the presence of the TTX-producing bacteria in non toxic newts from the current figures. Is there evidence for the absence of these 4 genera/11 strains in the non toxic population? The only comparison presented was made with OTUs from *Pseudomonas* (Figure 2C), but what about the OTUs for Aeromonas, Shewanella and Sphingopyxis?

3) Although the presence of TTX producing bacteria was shown by culturing the bacteria in vitro, both reviewers emphasized that no evidence is presented regarding their contribution to the presence of TTX on newt skin. Would it be possible to inoculate TTX producing bacteria to non toxic/skin sterilized newts and check for the production of TTX (complementation experiment, using toxic newts as positive controls)? Can this experiment be conducted within 2 months? in the current form, one can't exclude that the TTX producing bacteria strains have an inhibited production of TTX when present on the newt skin. In the case this concern can not be addressed, the authors should (i) calculate, based on the bacteria abundance data and the TTX production data, the overall expected production explained by those taxa compared to TTX production of toxic newts, (ii) down tone the conclusions of the document, (iii) revise/down tone the title and (iv) discuss this aspect in the paper.

4) The authors claim that TTX producing bacteria may drive an evolving arms race between prey and predators. This claim is supported by the fact that the skin microbiota is determined by some immune factors (eg genetic factors in newts). The argument would even be stronger if the bacteria were transmitted from one generation to the next (in/on the eggs?). Is there any information available on the topic? is there any available information regarding the origin of the bacteria the newts carry? Please discuss.

5) One reviewer mentioned that the hypotheses the authors wanted to test were difficult to grasp due to the overwhelming references to the Result section.

---

## [Author Response]

Essential revisions:1) The comparison between one toxic and one non toxic population of newts collected at different locations, but most importantly, at different time points (3 years apart and over different seasons) weakens the manuscript. Is there any evidence for the skin microbiome to be stable over such time/seasons that can be used to support this comparison? We believe that the manuscript is strong enough when focusing on toxic newts. We suggest the authors to move the data obtained about non toxic newts (half of Figure 1H and Figure 2) to the supplementary material. Please mention in the legend of these supplementary data that the 2 newt populations were collected at different time points.

In sampling and comparing the skin microbiota between toxic and non-toxic newts, we aimed to identify the bacterial types/OTUs present in each population while simultaneously sampling skin toxicity to determine if and how these two features might be related. Regardless of the temporal or geographical variables in this comparison, we believe the comparison is appropriate for our goal, which was simply to characterize the microbiota associated with newts that varied dramatically in toxicity. Thus, we have not moved the data from non-toxic newts to the supplement. However, we explicitly state that the samples were collected at different time points in the Results section.

Further, the long-term stability and effects of host species/identity and habitat on the cutaneous microbiota of wild amphibians is still a subject of ongoing research, and no clear associations can be made from existing data (see dois: 10.1038/ismej.2011.129, 10.1111/mec.12510, 10.3389/fmicb.2018.00442). We did not attempt to address questions of microbiome stability in this study, as our 16S-based characterization was intended to provide an initial insight into the skin-associated communities of wild newts and provide guidance for subsequent microbial cultivation in the lab. However, we do note that 8 of the 20 most abundant OTUs identified separately in each population are *shared* between the two populations. This information is included in Figure 1—source data 2. These OTUs are >97% identical in sequence and were identified in separate populations at separate times, which may be indicative of a core newt microbiome. However, we only briefly mention this observation (lines 158-161) and do not emphasize this point in the manuscript.

2) The comparison is also confusing as one can't extract the information regarding the presence of the TTX-producing bacteria in non toxic newts from the current figures. Is there evidence for the absence of these 4 genera/11 strains in the non toxic population? The only comparison presented was made with OTUs from Pseudomonas (Figure 2C), but what about the OTUs for Aeromonas, Shewanella and Sphingopyxis?

In the original manuscript, we focused on *Pseudomonas* because we found several strains that produce TTX. In addition, *Pseudomonas* were differentially abundant between toxic and non-toxic newts: our linear discriminant analysis effect size (LEfSe) revealed that *Pseudomonas* were among the top 10 most differentially abundant OTUs between the two populations overall and were statistically significantly more abundant in toxic newts than non-toxic newts (Fig 2D). Coupled with our cultivation data that revealed numerous TTX-producing *Pseudomonas* isolates, this led us to propose that toxic newts may harbor more TTX-producing *Pseudomonas* in their microbiota. This conclusion should be tested in future studies that sample the microbiota from numerous toxic and non-toxic newt populations to determine whether a true causal relationship exists; however, such work is outside the scope of our study.

In response to the reviewers’ question, we have added a heatmap of any OTU that was classified as *Pseudomonas, Aeromonas, Shewanella,* or *Sphingopyxis* as Figure 2—figure supplement 2. We found only a single OTU for *Sphingopyxis* and for *Shewanella*, both of which were only present in toxic newts and only in a few samples. The relative abundance of these OTUs was also low. We also identified three *Aeromonas* OTUs, one of which was highly abundant in all newt samples from both populations (Figure 2—figure supplement 2). This result was not surprising however, as *Aeromonas hydrophila* is a common commensal/opportunistic pathogen of amphibians and freshwater fishes (doi: 10.1016/j.micpath.2019.02.036). We added these findings to the discussion.

3) Although the presence of TTX producing bacteria was shown by culturing the bacteria in vitro, both reviewers emphasized that no evidence is presented regarding their contribution to the presence of TTX on newt skin. Would it be possible to inoculate TTX producing bacteria to non toxic/skin sterilized newts and check for the production of TTX (complementation experiment, using toxic newts as positive controls)? Can this experiment be conducted within 2 months? in the current form, one can't exclude that the TTX producing bacteria strains have an inhibited production of TTX when present on the newt skin. In the case this concern can not be addressed, the authors should (i) calculate, based on the bacteria abundance data and the TTX production data, the overall expected production explained by those taxa compared to TTX production of toxic newts, (ii) down tone the conclusions of the document, (iii) revise/down tone the title and (iv) discuss this aspect in the paper.

We understand the reviewers’ concern. We did try once to treat toxic newts with a broad-spectrum antibiotic. Overall, we did not observe significant changes in TTX levels at two weeks and one month following antibiotic treatment; we also found that the antibiotic did not substantially eliminate skin microbes. In addition, given that the newts sequester TTX in the granular glands of their skin, even killing all skin-associated bacteria may not substantially reduce newt toxicity.

Several previous studies have attempted to identify the source of TTX in rough-skinned newts and amphibians more broadly; however, we are the first to successfully isolate and culture TTX-producing bacteria from this species. Further, identifying the production of secondary metabolites from host-associated microbes is challenging when the researcher does not know whether host factors contribute to the production of the metabolite in vivo. Thus, the identification of TTX production in vitro is, we believe, a significant achievement.

To address another point the reviewers make, given the complications of host-symbiont interactions, attempting to calculate TTX production based on bacterial abundance data would almost certainly be inaccurate. Our in vitromeasurements of TTX production are likely an underestimate of the true biosynthetic potential of these isolated symbionts, particularly given that we cultured bacteria in nutrient-limited media. We have added these points to the discussion on lines 395-408. We feel that the alternative scenario presented by the reviewers, that “TTX producing bacteria strains have an inhibited production of TTX when present on the newt skin” is inconsistent with observations made in other host-bacterial interactions involving TTX. For relevant discussion of this topic, please see page 12 of doi: 10.3390/toxins9050166.

Finally, we have down toned several sentences (for example, lines 123-126) that were mentioned in the minor revisions below, but overall we feel that we have not overstated our results in the discussion. In the revised manuscript, the relevant paragraph (line 343 et seq.) begins “One of the most interesting insights to arise from this work is the possibility that the skin microbiome contributes to the predator-prey arms race between toxic newts and TTX-resistant garter snakes.” We feel this is an entirely appropriate conclusion based on our clear demonstration of TTX-producing bacteria isolated from the skin of toxic newts.

4) The authors claim that TTX producing bacteria may drive an evolving arms race between prey and predators. This claim is supported by the fact that the skin microbiota is determined by some immune factors (eg genetic factors in newts). The argument would even be stronger if the bacteria were transmitted from one generation to the next (in/on the eggs?). Is there any information available on the topic? is there any available information regarding the origin of the bacteria the newts carry? Please discuss.

The mode of transmission of the skin microbiota is not known for these newts, nor, to our knowledge, for any amphibian. Host-associated microbes may be acquired from the environment each generation, as in the bobtail squid, or they may be directly transmitted vertically within eggs, as is the case for aphids and many insect symbionts. The larger point is that TTX plays a well-documented, critical role in driving the arms race between newts and garter snakes (dois: 10.1007/s10886-005-1345-x, 10.1023/A:1021049125805, 10.1093/icb/43.3.408); our work adds an interesting angle to this classic body of work by showing that newts possess microbes that produce TTX, raising new questions about the target of selection in the newt-snake arms race. To clarify this point in the paper, we added a discussion of the heritability of host-associated bacteria in the discussion. We suspect that selection is acting at least in part on the newts, perhaps by increasing production of TTX precursors in newt skin or by changing the mucosal environment in a way that favors growth of TTX producers (e.g. antimicrobial peptide expression), but the list of potential targets is long and not terribly germane to the point of the paper.

5) One reviewer mentioned that the hypotheses the authors wanted to test were difficult to grasp due to the overwhelming references to the Result section.

We rewrote large sections of the introduction to clarify our hypotheses and experimental approach and removed extraneous details that are described later in the paper.